# mGRADE: Minimal Recurrent Gating Meets Delay Convolutions for Lightweight Sequence Modeling

## Abstract

Processing temporal data directly at the sensor source demands models that capture both short- and long-range dynamics under tight memory constraints. While State-of-the-Art (SotA) sequence models such as Transformers excel at these tasks, their quadratic memory scaling with sequence length makes them impractical for edge settings. Recurrent Neural Networks (RNNs) offer constant memory scaling, but train sequentially and slowly, and Temporal Convolutional Networks (TCNs), though efficiently trainable, also scale memory with kernel length. For more memory-efficient sequence modeling, we propose **mGRADE** (**m**inimally **G**ated **R**ecurrent **A**rchitecture with **D**elay **E**mbedding), a hybrid-memory system that integrates a temporal convolution with learnable spacings with a gated recurrent component equivalent to a minimal Gated Recurrent Unit (minGRU). The convolution with learnable spacings can express a flexible delay embedding that captures rapid temporal variations, while the recurrent component efficiently maintains global context with minimal memory overhead. We theoretically ground and empirically validate our approach on two types of synthetic tasks, demonstrating that mGRADE effectively separates and preserves temporal features across multiple timescales. Furthermore, on the challenging Long-Range Arena (LRA) benchmark as well as the 35-way Google Speech Commands (GSC) raw audio classification task, mGRADE reduces the memory footprint by up to a factor of 8, while maintaining competitive performance compared to SotA models.

## 1 Introduction

Embedded systems show great promise for temporal processing at the edge, enabling low-latency and energy-efficient inference for real-time tasks such as sensor data processing and autonomous control. However, the tight memory constraints of these systems make it difficult to process real-time streaming data, while also causally modeling dependencies across multiple timescales. Capturing long-range dependencies in sequence data requires storing information over long time horizons, whereas short-range dynamics demand high temporal resolution. Combining these multi-timescale dependencies strains the limited memory budgets of embedded systems, highlighting the need for more memory-efficient sequence models.

While State-of-the-Art (SotA) sequence models, such as Transformers (Vaswani et al., 2017) and Temporal Convolutional Networks (TCNs) (Waibel et al., 1989), can capture multi-timescale dependencies, they are ill-suited to embedded systems. This is mainly because their memory footprint grows with the sequence length, which impedes real-time processing of long sequences within a fixed memory size. In contrast, Recurrent Neural Networks (RNNs), particularly gated variants like the Long Short-Term Memory (LSTM) (Hochreiter & Schmidhuber, 1997; Gers et al., 2000) and the Gated Recurrent Unit (GRU) (Cho et al., 2014; Chung et al., 2014), offer constant inference-time memory over input sequences of arbitrary length. However, since they are not parallelizable, they are inefficient to train. Although linear time-invariant State-Space Models (SSMs) (Gu et al., 2022b) combine constant inference-time memory with efficient training, they lack selectivity over long-range dependencies (Gu & Dao, 2024) and require high precision parameters (Zhao et al., 2025), making them ill-suited for modeling multi-timescale data on limited-precision embedded systems.

Thus, despite the diversity of existing sequence models, they do not satisfy the tight memory requirements of embedded systems while efficiently capturing both long- and short-range dependencies required for advanced temporal signal processing. We address this challenge through three central contributions: (1) a novel sequence model architecture that integrates recurrent and modified convolutional components, (2) a theoretical analysis on why this integration excels at multi-timescale modeling under tight memory budgets, and (3) an empirical demonstration of our model's competitive performance and superior memory efficiency.

**Novel Architectural Integration** To meet the memory constraints of embedded systems while retaining the ability to simultaneously model short- and long-range dependencies, we introduce the **m**inimal **G**ated **R**ecurrent **A**rchitecture with **D**elay **E**mbeddings (mGRADE), a compact hybrid-memory sequence model combining a parallelizable gated recurrent component, the minimal Gated Recurrent Unit (minGRU) (Feng et al., 2025), with a causal temporal convolution featuring learnable spacings (Section 2). The modified convolution component replaces standard 1D-convolutions, enabling mGRADE to express a parameter-efficient delay embedding over the input within a single layer. This novel architecture achieves constant-time memory usage during inference, supports fully parallel training regardless of sequence length, and remains entirely causal, making it well-suited for real-time temporal processing in resource-constrained embedded environments.

**Theoretical Foundations of Hybrid Memory** We theoretically analyze how mGRADE's two components work together to provide complementary memory functions essential for efficient multi-timescale modeling. Through formal proofs and targeted synthetic tasks, we show that the learnable spacings in the convolution can express a parameter-efficient delay embedding, enabling generalization on fast, high-dimensional dynamics. Additionally, the minGRU-style gated recurrence achieves strong long-range modeling performance with high memory efficiency by compressing arbitrarily long histories into a fixed-size state. These findings provide a principled foundation for mGRADE's hybrid-memory architecture, where the learnable spacings capture short-range patterns with high parameter-efficiency, and the gated recurrence maintains long-range dependencies without increasing memory with sequence length.

**High Performance with High Memory Efficiency** Finally, we demonstrate that these theoretical capabilities translate to competitive sequence modeling capabilities at a far smaller memory cost than previously shown. Specifically, we benchmark mGRADE on the Long-Range Arena (LRA) tasks (Tay et al., 2021) and the 35-way Google Speech Commands (GSC) raw audio classification task (Warden, 2018), where we achieve competitive results with a memory footprint that is up to $8\times$ smaller than previously published SotA models.

By successfully modeling short- and long-range dependencies while satisfying tight memory constraints, mGRADE directly fills the critical gap in existing sequence models and enables advanced temporal signal processing in resource-constrained embedded systems.

## 2 MODEL ARCHITECTURE

The mGRADE architecture consists of an encoder (linear projection), a stack of $L$ mGRADE layers, and finally a decoder (non-linear projection) (Fig. 1A). The input sequence is streamed element by element, causally producing an output at every timestep $t$. The mGRADE layers are the core architectural feature, combining a modified depthwise 1D convolution with learnable spacings between kernel elements and a parallelizable gated recurrence, followed by an Multi-layer Perceptron (MLP) and layer normalization (Fig. 1B). Fig. 1C illustrates mGRADE's computational graph unrolled in time, demonstrating how the output at any given timestep $t$ depends on past and current inputs.

**Convolution with Learnable spacings** To enable mGRADE to capture short-term dynamics and high-frequency patterns (Section 3.1), we first pass the input to each mGRADE layer through a modified temporal convolution. To maximize expressivity without exploding the number of parameters, we learn the spacings between each kernel element using the Dilated Convolutions with Learnable Spacings (DCLS) framework from Hassani et al. (2023). This is equivalent to learning transmission delays over the input (Hammouamri et al., 2024). This choice is inspired by how tunable delays enrich the computational expressivity of spiking neural networks while simultaneously making them more parameter efficient. (Maass & Schmitt, 1999; D'agostino et al., 2024; Göltz et al., 2025).

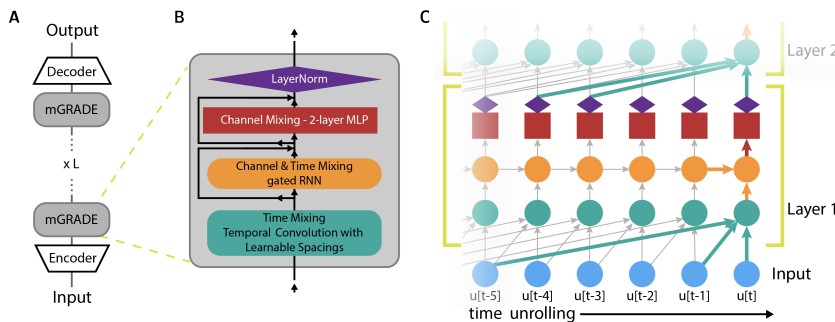

Figure 1: **Network architecture and spatio-temporal computational graph of mGRADE. A)** Network architecture composed of an encoder, $L$ mGRADE layers stacked on each other and a final decoder. **B)** A mGRADE layer is composed of four consecutive elements: a modified 1D convolution with learnable spacings, a gated RNN, a 2-layer MLP, and a layer normalization. It also employs skip connections around the gated RNN and the MLP. **C)** mGRADE's computational graph unrolled in time for the first two layers (time increasing from left to right). Bold colored arrows represent the flow of data that is being processed at timestep t. Light gray arrows represent past computations. Skip connections were omitted for simplicity.

Similar to classical TCNs, DCLS applies a discrete 1D convolution $x_{d,t} = (u_d * k_d)[t]$ over every channel $d \leq D$ of the input $\mathbf{u} \in \mathbb{R}^{D \times T}$ (with $T$ being the input sequence length). Unlike classical TCNs, the DCLS convolution kernel $k_d$ is parameterized by *two* sets $\Omega_d = \{w_0, w_1, ..., w_{K-1} \mid w_i \in \mathbb{R}\}$ and $\Psi_d = \{p_0, p_1, ..., p_{K-1} \mid p_i \in \mathbb{R}, \ p_i \leq p_{max}\}$ of $K$ elements each, representing the weights and positions in time of the trainable kernel elements. Each position in time is relative to the current timestep, making it equivalent to a transmission delay. The maximum position, $p_{max} = \Gamma$, defines the longest possible transmission delay applied to the input, thus indicating the total number of discrete timesteps that $k_d$ spans. Following Hassani et al. (2023), we will refer to $\Gamma$ as the *kernel length* and $K$ as the *kernel count*. Both kernel length and count are fixed across all channels.

To construct the discrete kernel $k_d$, each real-valued position $p_i$ is mapped to the discrete kernel indices $n \leq \Gamma$ via a differentiable interpolation function, $c$ (see Appendix A.1). This enables both the position and weight of the kernel elements to be learned with gradient descent. The kernel $k_d \in \mathbb{R}^{\Gamma}$ for a single channel then becomes:

$$k_d[n] = \sum_{i=0}^{K-1} w_i \cdot c[n, p_i], \quad \text{with} \quad k_d = [k_d[0], k_d[1], ..., k_d[\Gamma - 1]] \tag{1}$$

The DCLS convolution's output $x_{d,t}$ for each channel $d$ at timestep $t$ is then computed as a 1D convolution using the $k_d$ kernel over the past $\Gamma$ timesteps of the input. We stack all $D$ kernels $k_d$ into a kernel matrix $\mathbf{K} \in \mathbb{R}^{D \times \Gamma}$, yielding a final output vector $\mathbf{x}_t \in \mathbb{R}^D$ after the convolution.

**Gated recurrent component** To enable mGRADE to selectively model long-range dependencies (Section 3.2), we include a gated recurrent component in the mGRADE layer. To this end, we use a simplified and parallelizable version of the GRU (Cho et al., 2014), initially proposed by Martin & Cundy (2018) and also known as minGRU (Feng et al., 2025). minGRU removes the dependency of the update gate, $\mathbf{z}_t \in \mathbb{R}^H$, and candidate activation, $\tilde{\mathbf{h}}_t \in \mathbb{R}^H$, on the previous hidden state, $\mathbf{h}_{t-1} \in \mathbb{R}^H$. The hidden dimensionality $H$ is equal to $D$ times an expansion factor. Thus, given the output of the DCLS $\mathbf{x}_t \in \mathbb{R}^D$ and a linear activation on $\tilde{\mathbf{h}}_t$, $\mathbf{h}_t \in \mathbb{R}^H$ is updated as follows,

$$\mathbf{h}_t = (1 - \mathbf{z}_t) \odot \mathbf{h}_{t-1} + \mathbf{z}_t \odot \tilde{\mathbf{h}}_t \quad \text{with} \quad \mathbf{z}_t = \sigma(\mathbf{W}_z \mathbf{x}_t), \quad \tilde{\mathbf{h}}_t = \mathbf{W}_h \mathbf{x}_t, \tag{2}$$

where $\sigma$ is the sigmoid function, $\odot$ is the Hadamard product, and $\mathbf{W}_z$ and $\mathbf{W}_h \in \mathbb{R}^{D \times H}$ are the weights of the projections for $\mathbf{z}_t$ and $\tilde{\mathbf{h}}_t$, respectively.

The specific choice of a minGRU-style gated recurrence is motivated by its training efficiency and hardware compatibility. Since the update gate and candidate activation only depend on the current $\mathbf{x}_t$, the hidden states for every timestep can be computed in parallel using a prefix scan (Blelloch,

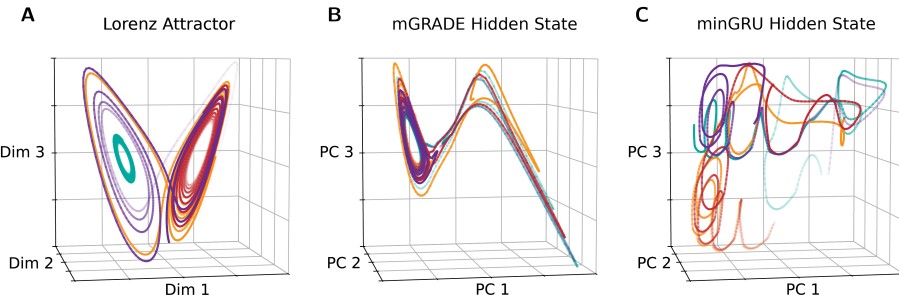

Figure 2: **mGRADE reconstructs a diffeomorphic mapping of the input dynamics. A)** Representative trajectories ($n = 4$) on the Lorenz attractor manifold with 5% Gaussian time-independent noise. The task is to predict dimension 1 at the next timestep. **B)** Representative trajectories in the hidden state space of a single-layer mGRADE projected to the first 3 Principal Components (PC). **C)** Trajectories in the hidden state space of a 2-layer minGRU projected to the first 3 Principal Components (PC). See Fig. A2 for all PCs compared individually.

1990), enabling efficient training in logarithmic time with respect to the sequence length (Feng et al., 2025). In addition, this architecture is well-suited to heavily quantized, low-power hardware implementations, as shown by (Billaudelle et al., 2025).

**MLP and Layer Normalization** Following the gated recurrent component, the hidden state $\mathbf{h}_t$ is passed through an MLP with $\mathbf{W}_{\text{MLP,in}} \in \mathbb{R}^{H \times 2D}$, $\mathbf{W}_{\text{MLP,out}} \in \mathbb{R}^{2D \times D}$, and a non-linearity in between. Afterwards, layer normalization is applied. Since $D$ represents the dimensionality of the activations passed between layers, we will call it the model dimensionality.

**Memory Complexity** During inference, mGRADE requires memory for both the model parameters and for the activation buffer, which stores all past and current activations needed to produce an output for the current timestep. The number of model parameters scales primarily with the model dimensionality $D$. Regarding the activation buffer, the gated recurrent component utilizes only a fixed-size hidden state vector. Thus, it can operate over arbitrary sequence lengths without scaling the activation buffer size, allowing us to fix the kernel length $\Gamma$ of the convolutional component while maintaining a theoretically unbounded temporal *receptive field*[1]. Accordingly, mGRADE's memory complexity is independent of the input sequence length over which it operates, in marked contrast to architectures like Transformers and TCNs, where memory requirements scale linearly with the sequence length or temporal receptive field. Further details can be found in Appendix A.2.

## 3  THEORETICAL FOUNDATIONS OF HYBRID MEMORY

We now develop a theoretical understanding of how mGRADE's learnable spacings and gated recurrent components complement each other. To this end, we first investigate how the learnable spacings enhance mGRADE beyond purely recurrent architectures by strengthening its structural inductive bias towards the reconstruction of short-term dynamics. We then show how the gated recurrent component enables long-range dependency learning by showing that a single mGRADE layer can formally model the Flip-Flop language and empirically solve the selective copying task, both of which require selectively remembering long-range dependencies that cannot be modeled by purely convolutional or non-gated recurrent architectures like TCNs or linear time-invariant SSMs.

### 3.1  LEARNABLE SPACINGS ENABLE SHORT-TERM PREDICTION OF DYNAMICS

mGRADE's temporal convolution with learnable spacings can be reframed as computing weighted sums of time-delayed inputs stored in cache memory at every timestep, with learnable positions controlling the durations of the delays (Hammouamri et al., 2024). This operation mirrors delay embeddings, a classical technique for time-series prediction and dynamical state-space reconstruction (Strogatz, 2015). Delay embeddings map an input sequence to a higher-dimensional vector

---

[1]range of past inputs that can influence the output at any given timestep $t$

Table 1: **Next-step prediction on 3D-Lorenz attractor.**

| Model | MASE (observed dim.) | MASE (unobserved dim.) | Near. Neigh. Overl. % | Params. |
|---|---|---|---|---|
| **mGRADE** | **0.38 ± 0.02** | **0.86 ± 0.11** | **32.7 ± 0.7** | **281** |
| minGRU | 0.63 ± 0.01 | 1.01 ± 0.01 | 28.8 ± 2.2 | 471 |

consisting of $m$ time-delayed copies of the original input. Takens' Embedding Theorem (Takens, 1981) guarantees that, for a $d$-dimensional dynamical system, any delay embedding of even a single observed dimension can diffeomorphically reconstruct the underlying manifold along which the system moves, using at most $m = 2d + 1$ delays in noise-free conditions. Intuitively, this means that given a vector with at least $m = 2d + 1$ different delays as input, the hidden state will trace out trajectories in $m$-dimensional space that resemble the underlying original dynamical system's trajectories – up to a smooth, invertible transformation.

**Theorem 1** (Informal). *A single-layer mGRADE can express a delay embedding of a $d$-dimensional dynamical system in the sense of Takens (1981), using only an $m$-dimensional projection of a single observed dimension as input. Its $m$-dimensional hidden state can thus learn to diffeomorphically reconstruct the system's full geometry over time.*

The full theorem and its proof are provided in Appendix B.1.1. It relies on the fact that mGRADE can learn distinct delays for each of the $m$ projections of the observed dimension, and then embed them directly into its hidden state. This internal representation captures the full geometry of the underlying dynamical system, allowing mGRADE to generalize to dimensions that were unobserved during training, specifically thanks to the learnable spacings.

We evaluate this claim on a next-step prediction task using the chaotic 3D-Lorenz attractor, training a single-layer mGRADE and a 2-layer minGRU[2] on 2000 noisy trajectories (Fig. 2A; Lorenz, 1963). To quantify the quality of the next-step predictions, we use the Mean Absolute Standardized Error (MASE) (Hyndman & Koehler, 2006). Note that MASE $> 1$ indicates that a model has no predictive power relative to naively predicting the current state's persistence (see Appendix B.1.2).

Visualizing the top three Principal Components of the 10 hidden states (Fig. 2B,C; Fig. A2), mGRADE's embedding reconstructs the Lorenz system's characteristic two-lobe structure, while the minGRU's embedding lacks similar visual correspondence. mGRADE also achieves a MASE that is $1.6\times$ lower than minGRU when predicting the next timestep on observed dimensions (Table 1; Fig. A1). When predicting the dimensions unobserved during training, mGRADE outperforms the 2-layer minGRU, which shows no predictive power given a MASE $> 1$. We also quantify how smoothly the geometry of the original attractor maps to the geometry of the hidden state space by measuring Nearest neighbor Overlap following (Ostrow et al., 2024) (for details see Appendix B.1.2). A high overlap percentage indicates that locally the two manifolds are smooth invertible mappings of each other, i.e., that the hidden space is a faithful diffeomorphic reconstruction of the original. Consistent with our visual check, mGRADE exceeds the minGRU by 4%.

This experiment highlights how adding the temporal convolution with learnable spacings, essentially a short-term cache of delayed inputs, implements a compact representation for reconstructing dynamical state spaces from partial, short-horizon observations. Importantly, the number of required delays (and thus parameters) grows linearly with the dimensionality of the underlying dynamical system rather than with sequence length, explaining mGRADE's parameter efficiency compared to models whose dimensionality needs to scale with sequence length (Schlag et al., 2021, Appendix A.2).

Additional experiments in Appendix B.3 show that the temporal convolution with learnable spacings also enables mGRADE to recognize and respond to high-frequency features far better than purely gated recurrent architectures, overcoming their bias towards low-frequency information (Rahaman et al., 2019).

---

[2]For a fair comparison, we give the minGRU an additional layer to provide its update gate and candidate activation in the second layer with temporal information. This means that the only significant difference between the mGRADE and minGRU models is the fact that mGRADE's uses a convolution.

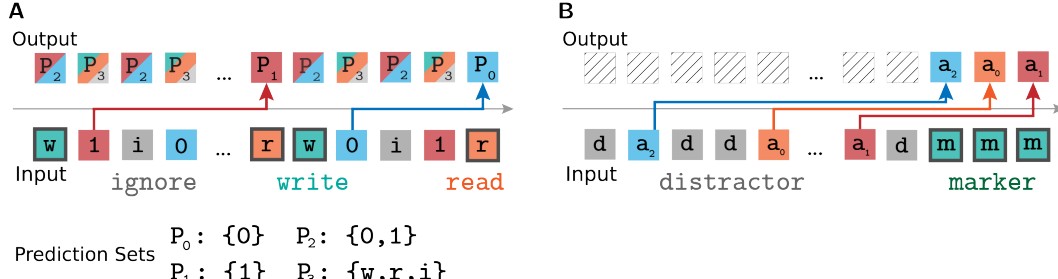

Prediction Sets
$P_0$: {0}   $P_2$: {0,1}
$P_1$: {1}   $P_3$: {w,r,i}

Figure 3: **Illustration of Flip-Flop and selective copying tasks. A)** Flip-Flop modeling consists of predicting the *prediction set* $P_i$ of next possible symbols in the given Flip-Flop string at every timestep. For r symbols, this is equivalent to recalling the value after the most recent w. **B)** Selective copying requires storing and recalling (after the marker symbol m) randomly distributed value symbols $a_i$ in the order they are presented in while ignoring the distractor symbols d in between.

## 3.2 GATED RECURRENCE ENABLES LONG-RANGE DEPENDENCY LEARNING

To probe mGRADE's ability to selectively remember long-range dependencies, we analyze its ability to predictively model Flip-Flop languages, a formal language family designed to test sequence models' long-range capabilities. (Fig. 3A; Liu et al., 2023).

**Definition 1** (Flip-Flop Language). Let the alphabet be $\Sigma = \{w, r, i, 0, 1\}$, where w, r, and i represent the instruction symbols for "write", "read", "ignore", and 0, 1 represent value symbols. Flip-Flop languages $L_{ff}$ consist of sets of strings over $\Sigma$ that alternate between instructions and values (e.g., w 0 r 0 i 1), satisfying the condition that after every r symbol, the subsequent symbol is equal to the value following the previous w. All valid strings begin with w.

**Definition 2** (Predictive Modeling). For a string $s \in L_{ff}$ and a prefix $s[1:t]$ ending at position $t$ with symbol $a_t$, predictive modeling requires outputting the *prediction set* $P_i \subseteq \Sigma$ of valid next symbols $a_{t+1}$ such that $s[1:t]\,a_{t+1}$ remains a prefix of a string in $L_{ff}$. We say that a model *predictively models* $L_{ff}$ iff its output at each timestep $t$ encodes all the information needed for a linear classifier to return the next prediction set with no errors.

In the Flip-Flop task, the model must predict at each timestep which class of next symbols can follow the current one such that the sequence remains a valid Flip-Flop string. For instance, when the model sees a w it should return the class including 0 *and* 1 (because both are valid next symbols), while after an r it depends on the value after the most recent w, which means either 0 *or* 1.

Predictive modeling of Flip-Flop languages is interesting for multiple reasons. First, a model's success on Flip-Flop modeling implies a broad computational expressivity on multiple formal languages and algorithmic simulation tasks (Liu et al., 2023). Second, Flip-Flop modeling requires maintaining the last w-paired value over arbitrarily long sequences. Accordingly, models with fixed-length context windows or sequence-length dependent memory scaling, such as TCNs and Transformers, cannot model Flip-Flop over arbitrary lengths with a fixed memory size (Sarrof et al., 2024; Liu et al., 2023, for proof see Appendix B.4.1). Finally, since any r is typically separated from the most recent w by an arbitrarily long string of irrelevant i-paired values, Flip-Flop modeling requires selectively remembering and ignoring value symbols based on the content of the preceding instruction. This content-aware *selectivity* across time factors into many real-world challenges, such as tracking filler-gap dependencies in natural language (Wilcox et al., 2018; Howitt et al., 2024) or ignoring irrelevant inputs during arithmetic reasoning (Shi et al., 2023). Notably, linear time-invariant SSMs without input-dependent gating, such as the Hungry Hungry Hippo (H3) (Dao et al., 2022) or Linear Recurrent Unit (LRU) (Orvieto et al., 2023), lack selectivity (Gu & Dao, 2024). In contrast to TCNs, Transformers, and linear time-invariant SSMs, a single-layer mGRADE can predictively model Flip-Flop languages due to its gated recurrent component.

**Theorem 2** (Flip-Flop Modeling with mGRADE). *A single-layer mGRADE with at least 2 delays can predictively model a Flip-Flop language, $L_{ff}$, at arbitrary length.*

*Proof Sketch.* (Full proof in Appendix B.4.1) mGRADE stores the value after the last w in one part of its hidden state, while the other merely reproduces the input. Learnable delays trigger the update

Table 2: **Flip-Flop and selective copying tasks.** The results on H3 (Dao et al., 2022) are from Gu & Dao (2024).

| | Flip-Flop | | | Selective Copying | | |
|---|---|---|---|---|---|---|
| | Test Acc. % | Params. | Buff. Activ. | Test Acc. % | Params. | Buff. Activ. |
| **mGRADE** | **99.6 ± 0.3** | **3K** | **96** | **87.1 ± 2.2** | **65K** | **17K** |
| LRU + DCLS | 88.5 ± 3.5 | 5K | 96 | 16.7 ± 3.5 | 81K | 17K |
| TCN | 60.6 ± 0.1 | 3K | 16K | 20.4 ± 0.1 | 80K | 806K |
| H3 (reproduction in Gu & Dao, 2024) | – | – | – | 57.0 | 166K | 512 |

gate of the storage hidden state only after a w (*selectively*), which then preserves the value over arbitrary sequence lengths via its recurrence until the next w. A linear classifier can then trivially extract the prediction set by reading the current input symbol from the reproducing hidden state, and, if the current input is r, reading out the stored value from the storage hidden state.

To evaluate this claim, we train a single-layer mGRADE, a single-layer linear time-invariant SSM, the LRU augmented with the DCLS convolution, and a 5-layer TCN on the Flip-Flop dataset from Liu et al. (2023). The training data consists of 1.6M Flip-Flop strings of 512 timesteps, where the expected distance between w and r is 10 timesteps. For testing, we used out-of-distribution data of the same length with sparse w and r (expected distance around 100 timesteps) to stress long-range dependency learning. Furthermore, we only report the recall accuracy, i.e., how often the model predicted the value following any given r symbol correctly. mGRADE solves the task to nearly 100%, substantially outperforming both the TCN and the SSM despite using less parameters and a smaller activation buffer, even at longer distances between successive w and r symbols (Fig. A5).

In addition to Flip-Flop language modeling, we further evaluate mGRADE's selectivity relative to linear time-invariant SSMs and TCNs by comparing the performance of mGRADE to LRU and a TCN on the well-established selective copying task (Fig. 3B; Jing et al., 2019; Gu & Dao, 2024). Selective copying is related to Flip-Flop modeling, but instead of providing an explicit instruction ahead of the sequence elements that contain relevant content (i.e. the w symbol used in $L_{ff}$), the content of the sequence element itself defines its relevance. Further details are in Appendix B.5.

Our results in Table 2 show that mGRADE clearly outperforms our LRU implementation by more than 70%, despite using $1.3\times$ less parameters. It also outperforms the TCN by more than 60% despite the TCN's receptive field being constructed to cover the entire input sequence (see Table A4). These results are consistent with the results achieved by Gu & Dao (2024) with H3 (Dao et al., 2022), another linear time-invariant SSM extending Structured State Space Model (S4) (Gu et al., 2022b). This emphasizes the importance of mGRADE's input-dependent gating over its hidden state transition, which both linear time-invariant SSMs and TCNs lack.

Overall, the Flip-Flop modeling and selective copying tasks confirm that mGRADE's minGRU-style recurrent component enable robust and selective long-range dependency modeling, outperforming purely convolutional and non-gated recurrent models without having to scale the memory size with input sequence length. These long-range learning capabilities combined with the ability of the learnable spacings to model short-term dynamics Section 3.1 underpin mGRADE's strong performance on real-world sequence tasks, as explored in the following section.

## 4 EMPIRICAL VERIFICATION

We test the proposed mGRADE architecture on two complementary sequence modeling benchmarks, LRA and GSC, designed to test its ability to handle both long- and short-range dependencies. The LRA benchmark (Tay et al., 2021) assesses the performance and inductive biases of mGRADE on long range dependency tasks across different modalities (text and flattened images), featuring sequences of lengths 1K to 8K[3]. The GSC task evaluates mGRADE on real-world time-series data, requiring the classification of raw speech recordings into one of 35 classes. Each audio sample is a one-second waveform recorded at 16 kHz, yielding sequences of length 16K. When evaluating

---

[3]Due to the compute-intensive nature of training on the PathX task in LRA, we leave this task for future work (see Appendix C). We note however, that the GSC raw-speech classification dataset consists of sequences with the same length as PathX (16K), hinting at mGRADE's potential for very long range dependency modeling.

the results, we do not solely focus on the achieved accuracy, but also consider the memory footprints of both parameters ("Params.") and activation buffer ("Buff."), which indicate how suitable the model is for deployment on embedded systems. The calculation of the activation buffer sizes is detailed in Appendix C.2, with Appendix C.3 outlining additional considerations regarding the actual instantiation on embedded platforms.

**Experimental setup** Since mGRADE is designed for real-time signal processing on embedded systems, it processes its inputs in a streamed and causal fashion. Therefore, we deliberately avoid using the (acausal) bidirectional processing used by S5 (Smith et al., 2023), S4-LegS (Gu et al., 2022a), and HGRN (Qin et al., 2023), although its inclusion leads to significant performance improvements on LRA, as shown by comparing S4-LegS to the causal S4 (Gu et al., 2022b) on Retrieval, Image, and Pathfinder (row 1 vs row 2 in Table 3). This pattern also holds for raw-speech classification, with (acausal) bidirectional models achieving a 3-5% gain (see top three rows of Table 4). Additionally, we process the raw inputs directly, consistent with all baselines, except for HGRN (Qin et al., 2023) which applies positional encoding to all LRA tasks except Pathfinder. Additional experimental details can be found in Appendix C.

**Results** In Table 3, we compare mGRADE's performance on LRA to current SotA RNN- and convolution-based architectures. Compared to the best-performing models, mGRADE reduces the memory footprint significantly, while still achieving comparable accuracies: for example, on ListOps, it achieves an accuracy within 0.9% of Liquid-S4's performance (Hasani et al., 2023), while using $7\times$ fewer parameters; on Pathfinder, it remains within 1.8% of MRConv-L's performance, while using $8\times$ smaller activation buffers. Compared to the models that are closest in size, mGRADE delivers higher performance: 1.9% higher accuracy than HGRN (Qin et al., 2023) on ListOps, 8% higher accuracy than S4 on Pathfinder (while still using $1.5$-$2\times$ fewer parameters). While the previous comparisons demonstrate that mGRADE attains comparable accuracy with substantially fewer parameters, an important complementary question is how the architectures behave under iso-parameter conditions. To assess this, we optimize an LRU model matched to mGRADE in both parameter count and activation footprint. Under this iso-capacity setting, mGRADE achieves an average 9% higher accuracy across all LRA tasks, indicating a clear architectural advantage beyond parameter efficiency alone. Note that unlike the results reported in (Orvieto et al., 2023), our LRU and mGRADE implementations are fully causal for every task, thus suitable for edge deployment.

In Table 4, we present the raw-speech classification results on GSC, comparing to current SotA recurrent architectures. mGRADE attains an accuracy within 2% of Liquid-S4 (Hasani et al., 2023) while requiring 10% fewer parameters. In addition, mGRADE relies solely on real-valued operations, avoiding the complex-valued arithmetic used in Liquid-S4. This combination of reduced parameter count and simpler computation makes mGRADE a suitable candidate for deployment under hardware or energy constraints.

These results confirm that our proposed architecture is capable of tackling large-scale and long-range tasks, particularly for time-series data, thus validating our theoretical predictions and demonstrating clear advantages in memory footprint and performance at comparable network sizes. We also assess whether these advantages carry over when comparing against architectures tailored for edge deployment, such as Spiking Neural Networkss (SNNs). As shown in Appendix D, mGRADE remains competitive in this regime, achieving accuracy within 2.6% of the strongest SNN while requiring $3\times$ less parameters.

To evaluate the impact of mGRADE's two architectural components on its performance, we also perform an ablation study in Appendix E.1. While the convolutions with learnable spacings and recurrent components perform well on their own on Image or ListOps, respectively, only full mGRADE can tackle *both* tasks successfully. Additionally, both components are needed to solve Pathfinder above chance level. Finally, we analyze the learned delays in Appendix E.2 and show that mGRADE flexibly learns task-specific convolution strategies, with spatially local information aggregation emerging in the learnable spacings of the convolution kernel for sequential image classification and distributed aggregation when detecting sparser structures.

Table 3: **Test accuracy on the LRA benchmark.** We differentiate the parameter count ("Params.") from the activation buffers ("Buff.") as explained in Appendix A.2. Parameter counts and accuracies not made available in the publications or official code are denoted by a dash. Best results are in bold, second best underlined.

| Model | ListOps | | Text | | Retrieval | | Image | | Pathfinder | |
|---|---|---|---|---|---|---|---|---|---|---|
| | Acc. | Params. / Buff. | Acc. | Params. / Buff. | Acc. | Params. / Buff. | Acc. | Params. / Buff. | Acc. | Params. / Buff. |
| *RNN-based architectures* | | | | | | | | | | |
| S4 (Gu et al., 2022b)[5] | 58.4 | 255K / 49K | 76.0 | 184k / 16K | 87.1 | 1.2M / 98K | 87.3 | 3.4M / 197K | 86.1 | 896K / 98K |
| S4-LegS (Gu et al., 2022a)[1,5] | 59.6 | 599K / 131K | 86.8 | 1.3M / 197K | 90.9 | 1.6M / 197K | 88.7 | 3.6M / 393K | 94.2 | 1.3M / 197K |
| DSS$_{SOFTMAX}$ (Gupta et al., 2022)[5] | 60.6 | 206K / 49K | 84.8 | 152K / 16K | 87.8 | 888K / 98K | 85.7 | 2.0M / 197K | 84.6 | **601K / 98K** |
| DSS$_{EXP}$[5] | 59.7 | 206K / 49K | 84.6 | 152K / 16K | 87.6 | 888K / 98K | 84.9 | 2.0M / 197K | 84.7 | **601K / 98K** |
| DSS$_{EXP-NO-SCALE}$[5] | 59.3 | 206K / 49K | 82.4 | 152K / 16K | 86.0 | 888K / 98K | 81.2 | 2.0M / 197K | 81.3 | **601K / 98K** |
| Liquid-S4 (Hasani et al., 2023)[5] | **62.8** | 333K / 8K | 89.0 | 164K / 4K | 91.2 | 1.5M / 98K | 89.5 | 11M / 1.6M | 94.8 | 1.2M / 98K |
| S5 (Smith et al., 2023)[1,5] | 62.2 | 190K / **0.1K** | 89.3 | 1.3M / 1.1K | 91.4 | 772K / 1.5K | 88.0 | 5.1M / **2.3K** | 95.3 | 1.1M / 1.5K |
| LRU (Orvieto et al., 2023)[3] | 60.2 | 190K / 1.5K | **89.4** | 1.3M / 1.1K | 89.9 | 772K / 1.5K | – | – / – | 95.1[1] | 1.1M / 1.5K |
| HGRN (Qin et al., 2023)[1,2,5] | 60.0 | 84K / 0.4K | 88.1 | 878K / 1.0K | **94.2** | 115K / **0.3K** | 88.7 | 20.6M / 6.1M | 92.9 | 1.3M / 1.5K |
| *Convolution-based architectures* | | | | | | | | | | |
| SGConv (Li et al., 2023)[4] | 61.5 | – / ∼ 1.5M | 89.2 | – / ∼ 1M | 91.1 | – / ∼ 6.3M | 88.0 | – / ∼ 3.1M | 95.5 | – / ∼ 1.6M |
| MRConv-L (Cunningham et al., 2024)[4] | 62.4 | 661K / ∼ 1.5M | 89.4 | – / ∼ 1M | 91.5 | – / ∼ 6.3M | 90.6 | 7.7M / ∼ 3.1M | 96.7 | – / ∼ 1.6M |
| *Our implementation* | | | | | | | | | | |
| LRU | 58.3 | 42K / 0.2K | 85.9 | 46K / **0.2K** | 86.8 | 118K / 0.4K | 84.3 | 1.2M / 2.3K | 57.4[6] | 1.1M / 1.5K |
| mGRADE | 61.9 | **40K** / 3K | 87.3 | **44K** / 1.5K | 88.1 | **104K** / 1.7K | 87.1 | **712K** / 197K | 94.9 | 612K / 197K |

[1] Bi-directional input processing.
[2] Uses positional encoding of the input.
[3] Assuming same hyperparameters as in S5 (Smith et al., 2023) as mentioned in (Orvieto et al., 2023) (official code not available).
[4] Buffer sizes calculated assuming same hyperparameters as in S4 (Gu et al., 2022b) as mentioned in (Li et al., 2023) and (Cunningham et al., 2024) (code not available).
[5] Parameter numbers extracted from the official GitHub repositories.
[6] Best validation accuracy using fully causal model (see Appendix C).

Table 4: **Test accuracy on the 35-way GSC classification task.** We differentiate the parameter count ("Params.") from the activation buffers ("Buff.") as explained in Appendix A.2. We differentiate between causal and bidirectional architectures as in (Gu et al., 2022a). Best results are in bold, second best underlined.

| Model | Parameters / Buff. | Causal | Bidirectional |
|---|---|---|---|
| S4-LegS (Gu et al., 2022a)[1,2] | 307K / 49K | 93.6 | 96.1 |
| S4-FouT (Gu et al., 2022a)[1,2] | 307K / 49K | 91.8 | 95.3 |
| S4D-LegS (Gu et al., 2022a)[1,2] | 306K / 49K | 93.6 | 95.8 |
| S4D-Inv (Gu et al., 2022a)[1,2] | 306K / 49K | 93.4 | 96.2 |
| S4D-Lin (Gu et al., 2022a)[1,2] | 306K / 49K | 93.4 | 96.3 |
| Liquid-S4 (Hasani et al., 2023)[1,2] | 224K / 5K | **96.8**[1] | - |
| S5 (Smith et al., 2023)[1,2] | 280K / **1K** | - | **96.5** |
| mGRADE | **198K** / 20K | 94.7 | - |

[1] Uses complex numbers in the recurrence.
[2] Parameter numbers extracted from the official publications.

## 5 RELATED WORKS

**Gated Recurrent Models** For many years, RNNs were virtually synonymous with sequence modeling. Gated RNNs, notably the LSTMs and GRUs, alleviate vanishing-gradient effects (Bengio et al., 1994; Hochreiter & Schmidhuber, 1997) through learned gating mechanisms that selectively control the flow of information and neuron state updates. Although effective, these models train sequentially and are therefore inefficient for training on very long sequences. Bradbury et al. (2017) and Martin & Cundy (2018) mitigate this bottleneck by removing the hidden-state dependency of the gate and candidate activation vectors, enabling parallel training through a prefix scan (Blelloch, 1990). In fact, the Gated Impulse Linear Recurrent (GILR) layer proposed by Martin & Cundy (2018) is equivalent to the minGRU-style gated recurrence used for mGRADE (see Section 2). Similar styles of recurrent gating have also been used in linear attention models, such as Gated Random Feature Attention (Peng et al., 2021) and Gated DeltaNet (Yang et al., 2025). Hierarchically Gated Recurrent Networks (HGRNs) adds complex-valued parameters and a hierarchical gating bias to a parallelizable gated RNN, setting an upper bound for the update gate and encouraging hierarchical processing of time scales from fast to slow (Qin et al., 2023). Note that this enforced low-frequency bias can impede recognizing important high-frequency features, which is already challenging for

gated RNNs (see Appendix B.3). mGRADE mitigates this by including the temporal convolution with learnable spacings in each layer.

**Linear Recurrent Models** Another line of research removes gating from the recurrent core altogether, favoring RNNs which rely on fully linear transitions between hidden states. Early investigations into linear RNNs (Mozer, 1993; Mikolov & Zweig, 2012; Pachitariu & Sahani, 2013) have converged in linear time-invariant SSMs, such as Structured State-Space Models (S4) (Gu et al., 2022b)), LRU (Orvieto et al., 2023), and the H3 architecture (Dao et al., 2022). Linear time-invariant SSMs are not *selective*, limiting their expressivity over long-range dependencies (see Section 3.2). Mamba (Gu & Dao, 2024) addresses this by reintroducing input-dependent update gating within the linear recurrence, effectively returning to the fold of parallelizable gated RNNs for large language modeling applications. While SSMs can be parallelized due to their linear recurrence, they use complex-valued parameters and highly specific initialization schemes, thus reducing their hardware compatibility and quantizability (Zhao et al., 2025). In contrast, the gated recurrent component of mGRADE has been successfully adapted for embedded deployment (Billaudelle et al., 2025).

**Temporal Convolution Models** TCNs were originally proposed in Waibel et al. (1989) to model temporal dependencies within a fixed-length receptive field using causal convolutions over the input sequence. Dilated convolutions, where the kernel elements are regularly spaced apart in time, efficiently expand the receptive field, particularly when these spacings increase exponentially with layer depth as in Wavenet (van den Oord et al., 2016). However, these fixed regular spacings can miss information in irregular frequencies, common in real-world signals (George & Smith, 1997). To address this, DCLS (Hassani et al., 2023) replaces fixed spacings with learnable ones, increasing performance and flexibility in temporal classification with SNNs (Hammouamri et al., 2024). However, DCLS as well as dilated TCNs still require buffering input activations, scaling memory cost with kernel length. This issue is exacerbated in global convolutional networks, such as SGConv (Li et al., 2023) and MRConv (Cunningham et al., 2024), where kernel length is matched to sequence length. Such approaches are impractical for embedded systems, given their memory constraints. mGRADE instead uses a fixed kernel length, resulting in a constant activation buffer size, relying on the recurrent component to capture long dependencies. Similar fixed-length convolutions have been successfully adapted to embedded systems, notably in DenRAM and Chameleon (D'agostino et al., 2024; den Blanken & Frenkel, 2025).

**Combining Convolutions and Recurrence** Several modern sequence models combine recurrent, convolutional, and normalization layers in multi-layered architectures. While this can improve performance, it complicates our understanding of each component's functional role. Also some attention-based models add a short 1-D convolution to their model backbone (Ma et al., 2023). Similar to mGRADE, recent gated recurrent architectures (Bradbury et al., 2017; Beck et al., 2024; Feng et al., 2025) as well as some linear RNNs (Dao et al., 2022) combine temporal convolutions and recurrent components, yielding consistent empirical performance improvements. However, the distinct functional contributions of these components, particularly with respect to their intrinsic timescales, have remained underexplored prior to this work. In addition, none of these works use learnable spacings which are critical for mGRADE's theoretical and empirical capabilities.

# 6 CONCLUSION

We present mGRADE, a hybrid-memory architecture engineered for real-time processing of sequences with multi-timescale dependencies on resource-constrained embedded systems. Our design is grounded in formal proofs and experimental evidence that demonstrate mGRADE's capacity to model not only short-term dynamics, but also long-range dependencies. We characterize the functional complementarity of each component: the learnable spacings in the temporal convolution serve as a short-term cache for delayed inputs, providing effective representations for reconstructing trajectories of dynamical systems from partial observations, while the gated recurrent component maintains a memory-efficient and selective history of the input sequence.

We support these theoretical arguments with an extensive empirical evaluation of mGRADE across the multi-timescale sequence modeling tasks of the LRA benchmark as well as the time-series raw audio classification GSC task. The results show that mGRADE substantially reduces the memory footprint compared to the SotA models, while maintaining competitive performance. This highlights the potential of mGRADE for large-scale, real-time sequence modeling on embedded systems.

## STATEMENT ON THE USAGE OF LLMS

In the preparation of this work, the authors used Large Language Models for the purpose of polishing and improving the readability of the text. This includes tasks such as correcting grammar and rephrasing sentences. After using the model, the authors reviewed and edited the content extensively and take full responsibility for all ideas, claims, and the final language presented in this paper.

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

# APPENDIX

## A  MODEL SPECIFICATION DETAILS

### A.1  LEARNABLE DCLS KERNELS

DCLS was first introduced by Hassani et al. (2023) and enables the spacings between different elements of a convolution kernel to be trained. In a temporal setting, this is equivalent to learning delays. DCLS requires a specific kernel construction parameterized by both a set of weights, $\Omega_d = \{w_0, w_1, ..., w_{K-1} \mid w_i \in \mathbb{R}\}$, and a set of corresponding positions, $\Psi_d = \{p_0, p_1, ..., p_{K-1} \mid p_i \in \mathbb{R}, \ p_i \leq p_{max}\}$, for every channel $d \leq D$ of the input. These sets have $K$, the kernel count, elements each, and a maximum possible position (or in time, a maximum delay) $p_{max} = \Gamma$, called the kernel length. In time, each position is relative to the current timestep, making it equivalent to a transmission delay.

To construct the discrete kernel $k_d$ for one of the input channels, each real-valued position $p_i$ is mapped to the discrete kernel indices $n \leq \Gamma$ via a differentiable interpolation function, $c$. This enables both the position and weight of the kernel elements to be learned with gradient descent. The kernel $k_d \in \mathbb{R}^\Gamma$ for a single channel then becomes:

$$k_d[n] = \sum_{i=0}^{K-1} w_i \cdot c[n, p_i], \quad \text{with} \quad k_d = [k_d[0], k_d[1], ..., k_d[\Gamma - 1]] \tag{3}$$

As in Khalfaoui-Hassani et al. (2023), we use a Gaussian with fixed width $v$ as our interpolation function:

$$c[n, p_i] = \exp\left[\frac{-1}{2}\left(\frac{n - p_i}{v}\right)^2\right] \tag{4}$$

The DCLS convolution's output $x_{d,t}$ for each channel $d$ at timestep $t$ is computed as follows:

$$x_{d,t} = (u_d * k_d)[t] = \sum_{n=0}^{\Gamma - 1} k_d[n] \odot u_d[t - n] \tag{5}$$

### A.2  MEMORY FOOTPRINT

**Scaling**  The memory requirements of mGRADE during inference consist of (1) the model parameters and (2) the activation buffer required for sequential processing. Since these come with distinct usage patterns and on-chip implementations in embedded systems, we treat them separately as they might necessitate employing different memory technologies.

In terms of the number of parameters, the temporal convolution component of each mGRADE layer scales with the number of channels (or model dimensionality) $D$ and the number of kernel elements $K$, leading to $\mathcal{O}(D \times K)$ complexity. In practice, $K$ is significantly lower than $D$ or $\Gamma$. The gated recurrent component scales with the model dimensionality and the hidden state dimensionality $H$, with $\mathcal{O}(D \times H)$, just like the MLP after the gated recurrence. Assuming that $H$ is proportional to $D$, the overall parameter memory scales as $\mathcal{O}(D^2)$.

In terms of activation buffer, the temporal convolution with learnable delays requires storing input activations for at most $\Gamma$ timesteps (Eq. (5)). More precisely, the activation buffer size scales linearly with the model dimensionality $D$, the number of layers $L$, and the kernel length $\Gamma$, yielding a memory complexity of $\mathcal{O}(D \times L \times \Gamma)$. The gated recurrent component only requires a single hidden state vector per layer (similar to the MLP), so assuming the hidden state dimensionality $H$ is proportional to $D$, the overall activation buffer complexity is thus dominated by the temporal convolution.

**Calculation**  This section presents complete derivations for the memory requirements of each network component during inference. We begin by examining the parameter memory footprint, followed by an analysis of buffer memory usage.

The notation MemParam$^{\text{component}}$ represents the memory consumption for each component (encoder, convolution, recurrent, MLP, decoder), with subelements categorized as Weights, Bias, and Positions. Note that the hidden state dimensionality of the recurrent component, $H$, is the model dimensionality scaled by an expansion factor denoted as $e$.

$$\text{MemParam}^{\text{Enc}} = \text{Weights}^{\text{Encoder}} + \text{Bias}^{\text{Encoder}} = D_{\text{in}} \times D \tag{6}$$

$$\text{MemParam}^{\text{Conv}} = \begin{cases} \text{Weights}^{\text{Conv}} + \text{Positions}^{\text{Conv}} = 2(K \times D) & \text{for mGRADE,} \\ \text{Weights}^{\text{Conv}} = K \times D & \text{else.} \end{cases} \tag{7}$$

$$\text{MemParam}^{\text{Rec}} = \text{Weights}_z + \text{Weights}_{\tilde{h}} + \text{Bias}_z + \text{Bias}_{\tilde{h}} \tag{8}$$

$$= H \times D + H \times D + H + H = 2eD^2 + 2D \tag{9}$$

$$\text{MemParam}^{\text{MLP}} = \text{Weights}^{\text{MLP}} + \text{Bias}^{MLP} \tag{10}$$

$$= 2D \times H + 2D + D \times 2D + D = 4eD^2 + 3D \tag{11}$$

$$\text{MemParam}^{\text{Norm}} = 2D \tag{12}$$

$$\text{MemParam}^{\text{Dec}} = \text{Weights}^{\text{Dec}} = D \times D_{\text{out}} \tag{13}$$

$$\text{MemParam}^{\text{network}} = \text{MemParam}^{\text{Enc}} + \text{MemParam}^{\text{Dec}}$$
$$+ L \times (\text{MemParam}^{\text{Conv}} + \text{MemParam}^{\text{Rec}} \tag{14}$$
$$+ \text{MemParam}^{\text{MLP}} + \text{MemParam}^{\text{Norm}})$$

Overall, this yields a memory complexity of $\mathcal{O}(L \times e \times D^2 + L \times K \times D)$ for parameter storage, with the model dimensionality $D$ dominating.

As explained in Section 2, any single convolution layer requires the storage of past inputs to produce an output. The past inputs are stored in an activation buffer whose size scales linearly with the kernel length $\Gamma$. Thus, for an entire network, the buffer requirements for the convolutional components are determined by $D$, $L$, and the kernel length $\Gamma$. In addition, recurrent models must maintain hidden state activations with dimensionality $H = eD$, further contributing to the required activation buffer.

$$\text{MemBuffer}^{\text{Conv}} = L \times D \times \Gamma \quad \text{with} \ \Gamma = \begin{cases} d \times (K-1) & \text{TCN with dilation rate } d, \\ p_{max} & \text{mGRADE.} \end{cases} \tag{15}$$

$$\text{MemBuffer}^{\text{Rec}} = L \times H = L \times eD \tag{16}$$

For purely recurrent models, the activation buffer size is determined entirely by MemBuffer$^{\text{Rec}}$ and, for purely convolutional models, by MemBuffer$^{\text{Conv}}$. mGRADE's final required memory is just the sum of the two:

$$\text{MemBuffer}^{\text{network}} = \text{MemBuffer}^{\text{Conv}} + \text{MemBuffer}^{\text{Rec}} \tag{17}$$

$$= L \times D \times \Gamma + L \times eD \tag{18}$$

$$= L \times D \times (\Gamma + e) \tag{19}$$

Thus, mGRADE's activation buffer does not scale with sequence length but instead with layer number, model dimensionality, and kernel length.

**Why not parameterize the convolution as a SSM?** Given the equivalence between convolutions and SSMs elaborated in Gu et al. (2022b), it is worth clarifying why we parameterize the learnable spacings with an explicit convolutional representation instead of a recurrent SSM. During training, when the full input sequence is available, the convolution can be efficiently computed via the FFT formulation. During inference, a fully instantiated convolution remains more memory-efficient. Representing an arbitrary convolution with an SSM during inference requires the same number of parameters and buffered activations as parameterizing the convolution directly. This is because, in

the worst case, an impulse response whose Hankel matrix has rank $\Gamma$ (equal to the kernel length) requires an SSM of state dimension $\Gamma$ (Schutter, 2000).

Thus, while it is theoretically possible to reformulate our convolution with learnable spacings as a fully recurrent SSM, doing so would not provide any memory advantage during inference or training. In fact, since we use learnable spacings between kernel elements, a naive SSM reformulation would require a transition matrix of size $\Gamma \times \Gamma$ even though only $K$ kernel weights are actually needed. For a discussion on the difference between *trainable* parameters and actual *instantiated* parameters during recurrent inference, see Appendix C.3.

## B  THEORETICAL CAPABILITIES OF MGRADE

### B.1  DYNAMICS RECONSTRUCTION TASK

#### B.1.1  PROOF FOR MGRADE AS A DELAY EMBEDDING

Here we detail the full proof of Theorem 1, demonstrating how mGRADE can learn to express a delay embedding.

**Theorem 1** (Reconstructing Dynamics through mGRADE's Delay Embedding). *Take a discrete-time dynamical system $f : \mathcal{M} \to \mathcal{M}$ over a compact manifold $M$ of dimension $d$, mapping $\mathbf{u}_t \in \mathbb{R}^d$ to $\mathbf{u}_{t+1} \in \mathbb{R}^d$ according to some differentiable and deterministic rule. Let $y : \mathcal{M} \to \mathbb{R}$ be a generic twice-differentiable observation function that deterministically maps any $\mathbf{u}_t$ on $\mathcal{M}$ to a single observable $y_t \in \mathbb{R}$ at time $t$. Let $m \geq 2d + 1$. Let a single mGRADE layer be preceded by a linear projection mapping the input $y_t$ to $\hat{\mathbf{y}}_t \in \mathbb{R}^m$ (the encoder) such that $D = H = m$. Assume that $v \to 0$, with $v$ being the width of mGRADE's interpolation function $c$.*

*Then, the hidden state $\mathbf{h}_t \in \mathbb{R}^m$ of $m$-dimensional mGRADE layer can learn to express a delay embedding in the sense of Takens (1981), and accordingly can learn to fully reconstruct the original dynamics of $\mathbf{u}_t \in \mathbb{R}^d$, differing only by a smooth, invertible change of coordinates (a diffeomorphism).*

*Proof.* Let $f : \mathcal{M} \to \mathcal{M}$ be a discrete-time dynamical system over a compact $d$-dimensional manifold $\mathcal{M}$, and let $y : \mathcal{M} \to \mathbb{R}$ be a generic twice-differentiable observation function, as defined in Theorem 3. Assume $m \geq 2d + 1$. Let the encoder preceding the mGRADE layer map $y_t$ to a $m$-dimensional vector $\hat{\mathbf{y}}_t = [y_t, y_t, \ldots, y_t] \in \mathbb{R}^m$ by replication.

Construct a single-layer mGRADE model as follows:

1. Each of the $m$ channels of mGRADE's temporal convolution kernel is learned to a unique, fixed delay. Specifically, for the $q$-th channel, set the kernel's delay to $p_q = (q - 1)\tau$, for some fixed delay step $\tau \in \mathbb{N}$, letting the interpolation function width $v \to 0$ and learning unitary weights. This means that the output of the $q$-th channel is equal to $y_{t-(q-1)\tau}$.

2. Let the update gate parameter matrix $\mathbf{W}_z$ be such that $\mathbf{h}_t = \tilde{\mathbf{h}}_t$ at every timestep (by setting $\mathbf{W}_z \to -\infty$). Let the hidden projection matrix $\mathbf{W}_h \in \mathbb{R}^{m \times m}$ be the identity.

Under this construction, the hidden state at each time $t$ becomes:

$$\mathbf{h}_t = \tilde{\mathbf{h}}_t = [y_t, y_{t-\tau}, y_{t-2\tau}, \ldots, y_{t-(m-1)\tau}],$$

which corresponds exactly to the delay vector used by Takens as the delay embedding (Takens, 1981).

Because $m \geq 2d + 1$ and $y$ is twice differentiable, Takens' Embedding Theorem guarantees that the map defined by mGRADE, $\mathbf{u}_t \mapsto \mathbf{h}_t$, is generically an embedding of the original dynamic manifold $\mathcal{M} \in \mathbb{R}^d$ into $\mathbb{R}^m$ (Takens, 1981). That is, the mGRADE hidden state $\mathbf{h}_t$ reconstructs the underlying system dynamics $\mathbf{u}_t$ up to a smooth, invertible change of coordinates (a diffeomorphism). $\square$

### B.1.2 TRAINING AND EVALUATION DETAILS

**Task Description**   We train mGRADE to perform autoregressive next-step prediction on the first dimension of 2000 randomly generated trajectories of 500 timesteps from the 3-dimensional chaotic Lorenz attractor, using the non-standardized Lorenz flow from the dysts package (Gilpin, 2024). We add 5% Gaussian time-independent observational noise to every trajectory. We test with randomly generated sequences of the same length as the training sequences, using either the first dimension or the 2 unobserved dimensions of the attractor (to evaluate generalization capabilities).

**Hyperparameters**   We compare a single-layer mGRADE to a 2-layer minGRU with the hyperparameters outlined in Table A1. We choose the 2-layer minGRU as a comparison for fairness, given that a single-layer minGRU does not provide its update gate or candidate activations with any temporal information (they only depend on the current input). Neither model uses the MLP or layer normalization. For the learning rate, we use AdamW (Loshchilov & Hutter, 2019) over all weights, and standard Adam (Kingma & Ba, 2017) for the biases, normalization layers, and DCLS positions. We use a cosine annealing learning rate scheduler without warmup. For the $\mathbf{z}$ gate biases, we use an "open" initialization for the (where the gate bias is set such that $\sigma(\mathbf{W}_z\mathbf{x}_t) \approx 1$) for all models, while using the traditional zero initialization for all other biases. For weights, we initialize with a truncated normal distribution, with a standard deviation set to $\sqrt{1/\text{fan\_in}}$. Other hyperparameters are outlined in Table A1. Reported results are averaged over 5 random seeds.

Table A1: Hyperparameters used for the Dynamics Reconstruction Task. L: number of layers. D/H: model dimensionality and hidden state size per layer (no hidden state expansion for any of these models). K: kernel count. Γ: kernel length. LR: learning rate. B: batch size. Epochs: max epochs set for the run. WD: weight decay.

| Parameter | L | D/H | K | Γ | LR | B | Epochs | WD |
|-----------|---|-----|---|---|-----|----|--------|-----|
| mGRADE | 1 | 10 | 1 | 32 | 0.004 | 32 | 200 | 0.0 |
| minGRU | 2 | 10 | – | – | 0.004 | 32 | 200 | 0.0 |

**Loss Metric**   Following (Ostrow et al., 2024), we use MASE as the loss metric. MASE compares the mean absolute error made by the model across a sequence with the mean absolute error that would have been incurred had the default prediction been that the next state is equal to the current state at every timestep (Hyndman & Koehler, 2006). This can be expressed as follows,

$$\text{MASE} = \frac{\frac{1}{T}\sum_{i=1}^{T}|y_t - \hat{y}_t|}{\frac{1}{T-1}\sum_{i=2}^{T}|y_t - y_{t-1}|},$$

where $\hat{y}$ is the model's prediction, $T$ is the sequence length, and $y_t$ is what is being predicted. Notably, a MASE > 1 means that naive forecasting (using the current $y_{t-1}$ to predict the next state $y_t$) works better than the forecasting model. Thus, a model only offers meaningful predictive power if it can achieve a MASE < 1.

**Manifold Similarity Metric**   The Nearest neighbor Overlap metric is used to evaluate how smoothly mGRADE maps the original Lorenz attractor manifold to the hidden state $h_t$. It is calculated following Ostrow et al. (2024). For every point $i$ on every trajectory in the test set, we find the k-nearest neighbors, i.e., the set of time points $Q_u(i)$ where the trajectory gets closest to the selected point in the original 3 dimensions of the Lorenz system. Then we map the selected point to the model's hidden state and evaluate how many of the k-nearest neighbors in the hidden state ($Q_h(i)$) are in fact the same k-nearest neighbors in the original system mapped into the hidden state. The number of overlapping neighbors relative to the total number of neighbors evaluated then yields the Nearest neighbor Overlap metric when applied to every point $i$ on every trajectory in the dataset:

$$\text{Overlap}(\textit{Original Manifold}, \textit{Hidden State}) = \frac{1}{n}\sum_{i=1}^{n}\frac{|Q_u(i) \cap Q_h(i)|}{k}$$

where $k$ is the number of neighbors evaluated (20 in this work), and $n$ is the total number of datapoints available across all trajectories.

**Results** The MASE over epochs for next-step prediction of the dimension that the models are trained on is shown for the mGRADE and minGRU models in Fig. A1A. Fig. A1B shows the average MASE when performing next-step prediction over the 2 dimensions that the models do not observe during training. mGRADE outperforms the purely recurrent models in both cases. Notably, the minGRU model does not generalize well to unobserved dimensions resulting in a MASE $> 1$. mGRADE also achieves a higher Nearest neighbor Overlap (in %) over epochs for both models in Fig. A1C.

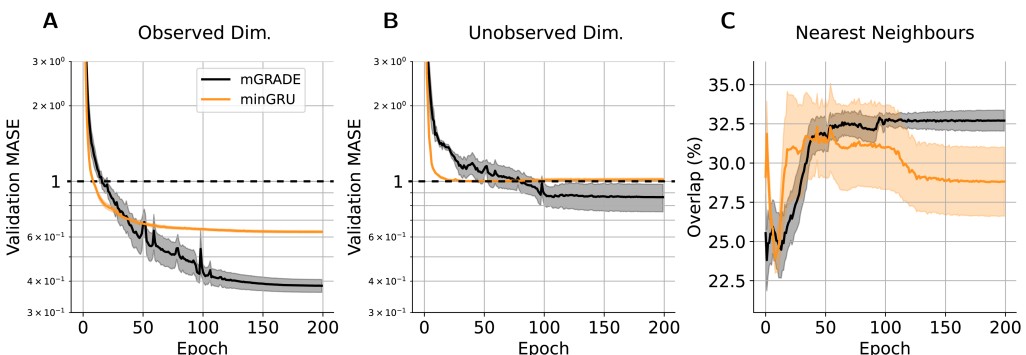

Figure A1: **mGRADE achieves lower training loss and higher manifold overlap, even on unobserved dimensions of dynamics. A)** Validation MASE loss (mean ± stde) over training epochs on predicting the observed first dimension. **B)** Out-of-Distribution MASE loss (mean ± stde) over training epochs on predicting the dimensions unobserved during training (2 and 3 of the original attractor). Dotted black line marks MASE $= 1$ which indicates no predictive power. **C)** Nearest neighbor Overlap (mean ± stde) of 20 nearest neighbors to each trajectory point between original state space and hidden state space over training epochs.

### B.2 HIDDEN STATE PRINCIPAL COMPONENTS

To provide an additional visualization aid on the similarity of the various models' hidden state representations of the Lorenz attractor dynamics in Section 3.1, we plot each of the top 3 Principal Components (PC) against each other in Fig. A2 together with their corresponding explained variance (EV). Compare to Fig. 2B, C for the corresponding 3D plots.

### B.3 HIGH-FREQUENCY PATTERN RECOGNITION TASK

**Task Description** We train on sequences with a total length of 165 timesteps, randomly generated at every training step. Each sequence contains 5 out of 16 possible features. Each feature is $l = 32$ timesteps long and contains input symbols selected from an alphabet of size $n = 16$. After each marker symbol m in the sequence, the goal is to output the associated class label $S_i$ of the preceding feature indexed by $i$. The *feature frequency* is adapted by changing how many times $r$ an input symbol is repeated within the feature before switching to a different input symbol (Figure A3). We normalize the feature frequency to the sampling rate so that it is equal to $1/r$. Thus, a feature frequency of 1.0 means that the input symbols in the pattern change every timestep, while a feature frequency of 0.5 means that each input symbol is repeated 2 times before switching. All models are trained on sequences that contain patterns with increasing normalized feature frequencies (0.07, 0.17, 0.33, 1.00).

**Hyperparameters** We compare single- and 2-layer mGRADEs with single- and 2-layer min-GRUs. The 2-layer models were chosen as comparisons to demonstrate that mGRADE can pass high-frequency feature information through multiple gated layers, while minGRU cannot. None of the models use an MLP or layer normalization between layers. For the optimization, we use AdamW (Loshchilov & Hutter, 2019) over weights, and standard Adam (Kingma & Ba, 2017) for the biases,

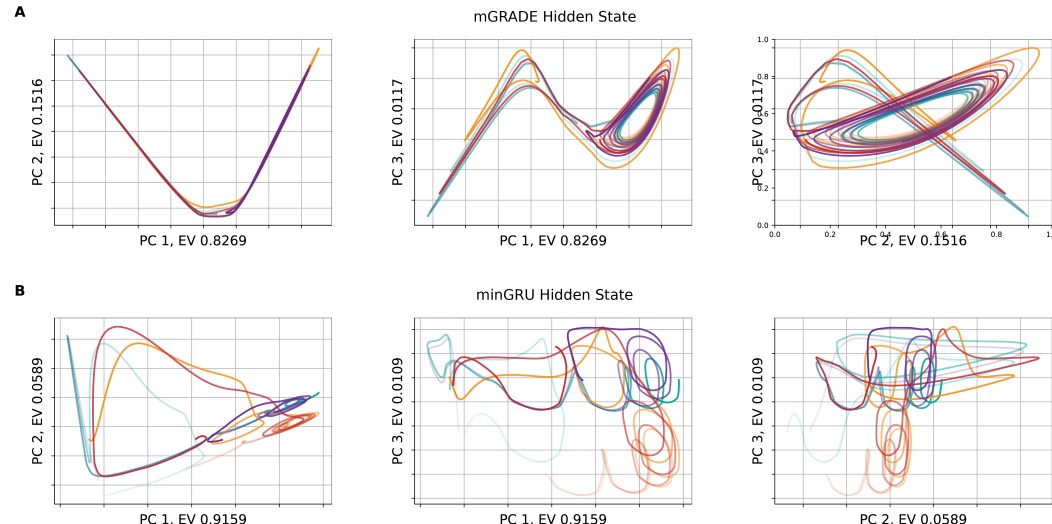

Figure A2: **mGRADE reconstructs original dynamics in hidden state principal components. A)** Three top PCs of single-layer mGRADE plotted against each other with corresponding explained variance (EV). (see Fig. 2B for 3D plot). **B)** Three top PCs of 2-layer minGRU plotted against each other with corresponding explained variance (EV). (see Fig. 2C for 3D plot).

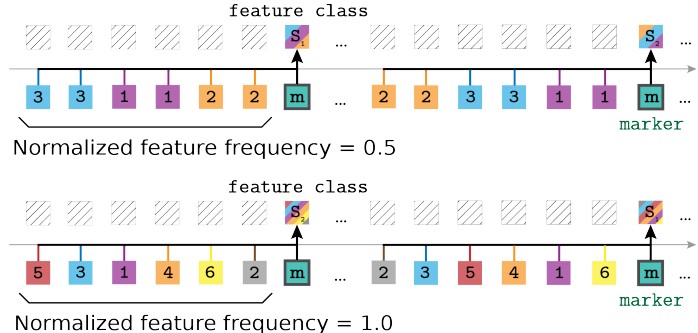

Figure A3: **High-frequency pattern recognition task.** The task requires classifying several features within a sequence, with each feature consisting of randomly ordered input symbols with different feature frequencies (top: low frequency, bottom: high frequency). After being presented a feature, the model should output the associated class. The number of times that the input symbol changes within a feature is the inverse of the feature frequency.

normalization layers, and DCLS positions. We use a cosine annealing learning rate scheduler without warmup. For biases, we use a standard zero initialization, and for weights, we initialize with a truncated normal distribution, with a standard deviation set to $\sqrt{1/\text{fan\_in}}$. Other hyperparameters are outlined in Table A2. Reported results are averaged over 5 random seeds.

**Loss Metric**    Cross-entropy loss is used over the predicted feature classes, which are set for each timestep with a marker $m$ in the input. Each model is tested on randomly generated test sequences using the same feature frequency as during training.

**Results**    Fig. A4A shows the classification validation accuracy over training steps for the models that were trained on a feature frequency of 0.33. With increasing feature frequency, the test accuracy of the mGRADE models remains roughly constant (or even slightly increases) while the 2-layer minGRU collapses when trying to classify higher-frequency features (Fig. A4B). Notably, adding a second layer to the minGRU decreases its classification accuracy, implying that high-frequency information is lost over successive gated recurrent layers as suggested by previous literature (Rahaman

Table A2: Hyperparameters for the High Frequency Recognition Task. L: number of layers. D/H: model dimensionality and hidden state size per layer (no hidden state expansion for any of these models). K: kernel count. $\Gamma$: kernel length. LR: learning rate. B: batch size. Training Steps: number of batches presented during training. WD: weight decay.

| Parameter | L | D/H | K | $\Gamma$ | LR | B | Training Steps | WD |
|---|---|---|---|---|---|---|---|---|
| mGRADE | 1 | 16 | 8 | 16 | 0.004 | 64 | 200 | 0.1 |
| mGRADE | 2 | 16 | 8 | 16 | 0.004 | 64 | 200 | 0.1 |
| minGRU | 1 | 20 | – | – | 0.004 | 64 | 200 | 0.1 |
| minGRU | 2 | 16 | – | – | 0.004 | 64 | 200 | 0.1 |

et al., 2019). mGRADE on the other hand maintains performance even with 2 layers, indicating that the temporal convolution indeed preserves high-frequency information even through the gated recurrent unit.

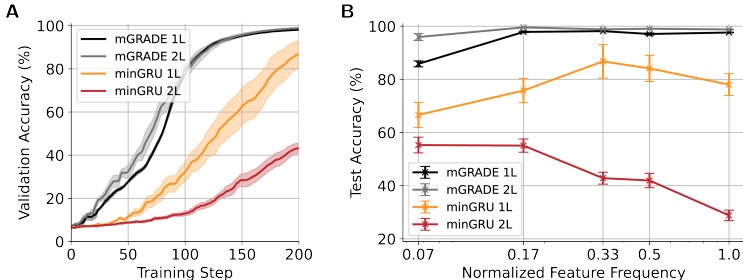

Figure A4: **mGRADE recognizes features with high frequencies better than pure gated RNNs. A)** Validation accuracy (mean ± stde) for high-frequency pattern recognition task with a feature frequency of 0.33. **B)** Final test accuracy (mean ± stde) after training on different feature frequencies. 1L stands for single-layer and 2L for 2-layer models.

### B.4 FLIP-FLOP PREDICTIVE MODELING TASK

#### B.4.1 PROOFS FOR FLIP-FLOP PREDICTIVE MODELING CAPABILITIES

Here we detail the full proofs associated with Section 3.2. For convenience, we start by reiterating the definition of a Flip-Flop language.

**Definition 1** (Flip-Flop Language). Let the alphabet be $\Sigma = \{w, r, i, 0, 1\}$, where $w, r$, and $i$ represent instruction symbols ("write", "read", "ignore"), and $0, 1$ represent value symbols. The *Flip-Flop languages* $L_{ff}$ consist of sets of strings over $\Sigma$ that alternate between instructions and values (e.g., w 0 r 0 i 1), satisfying the condition that after every r symbol, the subsequent value equals the value symbol following the most recent w. All valid strings begin with w.

**Definition 2** (Predictive Modeling). For a string $s \in L_{ff}$ and a prefix $s[1 : t]$ ending at position $t$ with symbol $a_t$, predictive modeling requires outputting the **prediction set** $P_i \subseteq \Sigma$ of valid next symbols $a_{t+1}$ such that $s[1 : t] a_{t+1}$ remains a prefix of some string in $L_{ff}$. We say that a model **predictively models** $L_{ff}$ iff its output at a single timestep $t$ encodes all the information needed such that a linear classifier can return the next prediction set with $100\%$ accuracy.

**mGRADE**

**Theorem 2** (Flip-Flop Modeling with mGRADE). *A single-layer mGRADE with at least 2 delays in the convolutional component can predictively model a Flip-Flop language, $L_{ff}$, at arbitrary length.*

*Proof.* We prove the above theorem by construction. For notation, we use to $\mathbf{c} = [\mathbf{a}, \mathbf{b}]$ to denote stacking column vectors $\mathbf{a} \in \mathbb{R}^A$, $\mathbf{b} \in \mathbb{R}^B$ into another column vector $\mathbf{c} \in \mathbb{R}^{A+B}$. In addition,

we assume all symbols and possible prediction sets are one-hot encoded in the input and output, respectively.

Consider a single-layer mGRADE with an input sequence of length $T$, $\mathbf{u}_{1:T} \in \mathbb{R}^{|\Sigma| \times T}$, where $\mathbf{u}_t$ is the one-hot encoded vector of the symbol at position $t$ in the string, and a model dimensionality of $D = 2|\Sigma|$. To match the model dimensionality, pass $\mathbf{u}_{1:T}$ at every timestep through a simple linear projection to match the model dimensionality (as described in the model architecture). Set this linear projection to simply stack 2 copies of the input $\mathbf{u}_{1:T}$ in a single vector $\hat{\mathbf{u}}_{1:T} = [\mathbf{u}_{1:T}, \mathbf{u}_{1:T}]$. Set the convolution component to have 2 different delays to the 2 copies of the input in $\hat{\mathbf{u}}_{1:T}$ at times $t$ and $t-1$ (delay of 0 and 1 respectively) with weights of 1. Set the interpolation function width $v$ narrow enough such that the convolution kernel elements are zero everywhere besides at $t$ and $t-1$. Given this kernel construction, the output of the temporal convolution at any timestep $\mathbf{x}_t \in \mathbb{R}^{2|\Sigma|}$ will depend only on the current input $\mathbf{u}_t, \mathbf{u}_{t-1} \in \mathbb{R}^{|\Sigma|}$. Specifically, given the 2 different delays to the input copies, $\mathbf{x}_t = [\mathbf{u}_t, \mathbf{u}_{t-1}]$.

Set the hidden state size $H = D = 2|\Sigma|$ such that $\mathbf{h}_t \in \mathbb{R}^{2|\Sigma|}$. Split the hidden state into 2 components $\mathbf{h}_t = [\mathbf{h}_t^{\text{stored}}, \mathbf{h}_t^{\text{current}}]$ where $\mathbf{h}_t^{\text{stored}} \in \mathbb{R}^{|\Sigma|}$ is parameterized such that it stores the value following the most recent $\mathtt{w}$, and $\mathbf{h}_t^{\text{current}} \in \mathbb{R}^{|\Sigma|}$ passes on the current input. Define $\mathbf{z}_t = [\mathbf{z}_t^{\text{stored}}, \mathbf{z}_t^{\text{current}}]$, and $\tilde{\mathbf{h}}_t = [\tilde{\mathbf{h}}_t^{\text{stored}}, \tilde{\mathbf{h}}_t^{\text{current}}]$ as the corresponding gate and candidate states.

The mGRADE updates $\mathbf{h}_t$ as follows.

1. Compute $\mathbf{z}_t^{\text{stored}}$:

$$\mathbf{z}_t^{\text{stored}} = \sigma(\mathbf{W}_z^{\text{stored}} \mathbf{x}_t) = \sigma(\mathbf{W}_z^{\text{stored}}[\mathbf{u}_t, \mathbf{u}_{t-1}])$$

where $\sigma$ is the sigmoid function, and $\mathbf{W}_z^{\text{stored}} \in \mathbb{R}^{|\Sigma| \times 2|\Sigma|}$ is the weight matrix coupling $\mathbf{z}_t^{\text{stored}}$ and $\mathbf{x}_t = [\mathbf{u}_t, \mathbf{u}_{t-1}]$. Set $\mathbf{W}_z^{\text{stored}}$ such that the weight corresponding to the location of the 1 in the one-hot encoding of $\mathtt{w}$ in $\mathbf{u}_{t-1}$ approaches $-\infty$, and weights for all other components to approach $+\infty$. Thus:

$$\mathbf{z}_t^{\text{stored}} = \begin{cases} 0 & \text{if } \mathbf{u}_{t-1} = \mathtt{w}, \\ 1 & \text{otherwise.} \end{cases}$$

2. Compute $\tilde{\mathbf{h}}_t^{\text{stored}}$:

$$\tilde{\mathbf{h}}_t^{\text{stored}} = \mathbf{W}_{\tilde{h}}^{\text{stored}} \mathbf{x}_t = \mathbf{W}_{\tilde{h}}^{\text{stored}}[\mathbf{u}_t, \mathbf{u}_{t-1}]$$

where $\mathbf{W}_{\tilde{h}}^{\text{stored}} \in \mathbb{R}^{|\Sigma| \times 2|\Sigma|}$ is the weight matrix coupling $\tilde{\mathbf{h}}_t^{\text{stored}}$ and $\mathbf{x}_t = [\mathbf{u}_t, \mathbf{u}_{t-1}]$. Set $\mathbf{W}_{\tilde{h}}^{\text{stored}}$ as a block matrix containing the identity in the component multiplied with $\mathbf{u}_t$ and zeros in the component multiplied with $\mathbf{u}_{t-1}$ to the effect that the $\mathbf{u}_t$ component gets passed on whereas $\mathbf{u}_{t-1}$ does not. Thus:

$$\tilde{\mathbf{h}}_t^{\text{stored}} = \mathbf{u}_t$$

3. Update $\mathbf{h}_t^{\text{stored}}$:

$$\mathbf{h}_t^{\text{stored}} = \mathbf{z}_t^{\text{stored}} \odot \mathbf{h}_{t-1}^{\text{stored}} + (1 - \mathbf{z}_t^{\text{stored}}) \odot \tilde{\mathbf{h}}_t^{\text{stored}}$$

When $\mathbf{u}_{t-1} = \mathtt{w}$, $\mathbf{z}_t = 0$ in the asymptotic limit, so $\mathbf{h}_t^{\text{stored}} = \tilde{\mathbf{h}}_t^{\text{stored}} = \mathbf{u}_t$ (a value $\mathtt{0}$ or $\mathtt{1}$); otherwise, $\mathbf{z}_t = 1$, so $\mathbf{h}_t^{\text{stored}} = \mathbf{h}_{t-1}^{\text{stored}}$.

4. Update $\mathbf{h}_t^{\text{current}}$. Let each component of $\mathbf{W}_z^{\text{current}} \in \mathbb{R}^{|\Sigma| \times 2|\Sigma|}$ approach $-\infty$ such that $\mathbf{z}_t^{\text{current}} \approx 0$ always. As above, set $\mathbf{W}_{\tilde{h}}^{\text{current}} \in \mathbb{R}^{|\Sigma| \times 2|\Sigma|}$ such that $\tilde{\mathbf{h}}_t^{\text{stored}} = \mathbf{u}_t$. In the asymptotic limit, the update expression

$$\mathbf{h}_t^{\text{current}} = \mathbf{z}_t^{\text{current}} \odot \mathbf{h}_{t-1}^{\text{current}} + (1 - \mathbf{z}_t^{\text{current}}) \odot \tilde{\mathbf{h}}_t^{\text{current}}$$

evaluates to $\mathbf{h}_t^{\text{current}} = \mathbf{u}_t$.

Thus, $\mathbf{h}_t^{\text{stored}}$ retains the one-hot vector of the value following the most recent $\mathtt{w}$, and $\mathbf{h}_t^{\text{current}}$ passes on $\mathbf{u}_t$.

The possible prediction sets over $L_{ff}$ are $P_1 = \{\mathtt{0}, \mathtt{1}\}$, $P_2 = \{\mathtt{w}, \mathtt{r}, \mathtt{i}\}$, $P_3 = \{\mathtt{0}\}$, $P_4 = \{\mathtt{1}\}$. Given the structure of $L_{ff}$, we can associate each set of possible input symbols to its corresponding

prediction set.

If $\mathbf{u}_t \in \{0, 1\}$ (a value), then $\mathbf{u}_{t+1}$ is an instruction, so the output should be $P_2 = \{w, r, i\}$. If $\mathbf{u}_t \in \{w, r, i\}$ (an instruction), then $\mathbf{u}_{t+1}$ must be a value. Specifically, if $\mathbf{u}_t = w$ or $\mathbf{u}_t = i$, then the output should be $P_1 = \{0, 1\}$, since the following value is arbitrary. On the other hand, if $\mathbf{u}_t = r$, then $\mathbf{u}_{t+1}$ must match the value after the most recent $w$, which (by the preceding construction) is stored in $\mathbf{h}_t^{\text{stored}}$. Thus, the output at time $t$ should be

$$\mathbf{h}_t^{\text{stored}} = \begin{cases} p_3 & \text{if } \mathbf{h}_t^{\text{stored}} = 0, \\ p_4 & \text{otherwise.} \end{cases}$$

.

A linear classifier, parameterized by a weight matrix $\mathbf{W}_c \in \mathbb{R}^{4 \times 2|\Sigma|}$ and bias $\mathbf{b}_c \in \mathbb{R}^4$, maps $\mathbf{h}_t$ to the correct prediction set $P_i$:

$$i = \text{argmax}(\mathbf{W}_c \mathbf{h}_t + \mathbf{b}_c),$$

where $i$ is the index corresponding to one of the four prediction sets. Since $\mathbf{h}_t = [\mathbf{h}_t^{\text{stored}}, \mathbf{u}_t]$ provides both the current symbol and the stored value, $\mathbf{W}_c$ can be trained (or constructed) to distinguish these cases based on the one-hot encoded positions in $\mathbf{h}_t^{\text{current}} = \mathbf{u}_t$ and $\mathbf{h}_t^{\text{stored}}$. Specifically, we set $\mathbf{W}_c$ to have a high weight between the one-hot encodings of $w$ and $i$ and the corresponding index of the prediction set $\{0,1\}$. We also set a high weight between the one-hot encodings of $0$ and $1$ and the corresponding index of the prediction set $\{w, r, i\}$. Finally, we set a negative bias term to the indices of the $\{0\}$ and $\{1\}$ prediction sets with a corresponding larger weight between $r$ and those indices such that $r$ is enough to activate $\{0\}$ and almost enough to activate $\{1\}$. If now the weight from $\mathbf{h}_t^{\text{stored}}$ to $\{0\}$ is negative and to $\{1\}$ is positive, then a stored $1$ will activate $\{1\}$ (and suppress $\{0\}$) while a stored $0$ will allow $\{0\}$) to be active while not adding to the activation of $\{1\}$.

To conclude, for any prefix $s[1:t] \in L_{ff}$:

1. $\mathbf{h}_t^{\text{stored}}$ can correctly store the value symbol after the most recent $w$ (at least in an asymptotic limit w.r.t. the weight magnitudes of the update gate $\mathbf{z}_t^{\text{stored}}$).

2. $\mathbf{h}_t^{\text{current}} = \mathbf{u}_t$ encodes the current input symbol.

3. Given mGRADE's outputs $\mathbf{h}_t$, a linear classifier can be constructed to output the correct prediction set classification as required by the language's rules, handling arbitrary lengths since $\mathbf{h}_t^{\text{stored}}$ persists across timesteps.

For initial states or prefixes without $w$, assume $\mathbf{h}_0^{\text{stored}} = \mathbf{0}$, but since every $r$ in a valid string follows a $w$, $\mathbf{h}_t^{\text{stored}}$ is always defined when needed. Thus, mGRADE can predictively model $L_{ff}$ at arbitrary length. □

### Fixed-length Context Models

**Theorem 3** (Flip-Flop Modeling with Fixed-length Context Models). *A model with a fixed-length context window for a fixed memory size cannot predictively model a Flip-Flop language, $L_{ff}$, at arbitrary lengths.*

*Proof.* Consider a sequence of length $T_c + 3$, where $T_c$ is the context window given some fixed memory size. Start with $w\ v$, follow with $T_c\ i$ instructions, and end with $r$. The correct prediction after $r$ is $v$, but $v$ lies outside the context window, forcing chance-level performance. Note that increasing the context length is the obvious solution to this problem however, the correspondingly increasing memory costs eventually become prohibitive for very long sequences. □

Note that models with fixed-length temporal contexts given some fixed memory size include Transformers and TCNs.

### B.4.2 TRAINING AND EVALUATION DETAILS

**Task Description** We train on the Flip-Flop dataset from Liu et al. (2023), containing 1M valid Flip-Flop sequences of 512 timesteps, where training data contains $i$ instruction symbols with probability $p(i) = 0.8$, such that the expected distance between any $w$ and $r$ symbol is 10 timesteps.

This dataset slightly simplifies the full predictive modeling task to focus on recalling the correct value symbol after an `r` as described in Section 3.2. For testing, we use out-of-distribution data with sparse `w` and `r` (expected distance around 100 timesteps) to stress long-range dependencies.

**Hyperparameters**  We compare a single-layer mGRADE to a 5-layer TCN (with an exponentially increasing receptive field) and a single-layer LRU augmented by DCLS. The LRU is used as a drop-in replacement for the gated recurrent component of mGRADE, demonstrating the importance of mGRADE's update gate relative to a linear time-invariant SSM like the LRU. We use the code supplied in (Zucchet, 2024) as the basis for our LRU implementation. None of the models use layer normalization. For the optimization, we use AdamW (Loshchilov & Hutter, 2019) over the weights, and Adam (Kingma & Ba, 2017) for the biases, the normalization layers, and the DCLS positions. We also use a cosine annealing learning rate scheduler without warmup. For the $\mathbf{z}$ gate biases, we use a "closed" initialization (where the gate bias is set such that $\sigma(\mathbf{W}_z\mathbf{x}_t) \approx 0$), while using the traditional zero initialization for all other biases. For all weights, we initialize with a truncated normal distribution with a standard deviation set to $\sqrt{1/\text{fan\_in}}$. Other hyperparameters are outlined in Table A3. Reported results are averaged over 3 random seeds.

Table A3: Hyperparameters for the Flip-Flop Modeling Task. L: number of layers. D/H: model dimensionality and hidden state size per layer (no hidden state expansion for any of these models). K: kernel count (for all layers). $\Gamma$: kernel length (per layer from input to output for the TCN). LR: learning rate. B: batch size. Training Steps: number of batches presented during training. WD: weight decay.

| Parameter | L | D/H | K | $\Gamma$ | LR | B | Training Steps | WD |
|---|---|---|---|---|---|---|---|---|
| mGRADE | 1 | 32 | 1 | 2 | 0.004 | 64 | 250,000 | 0.1 |
| LRU + DCLS | 1 | 32 | 1 | 2 | 0.004 | 64 | 250,000 | 0.1 |
| TCN | 5 | 32 | 16 | 16/32/64/128/256 | 0.004 | 64 | 250,000 | 0.1 |

**Loss Metric**  Cross-entropy loss is used as the training loss. The reported accuracy is how often the model correctly recalls the value symbol after the most recent `w` when encountering a `r`. Since there are 2 possible value symbols, chance level performance lies at 50%.

**Results**  Fig. A5A shows the recall accuracy of each model over training steps.

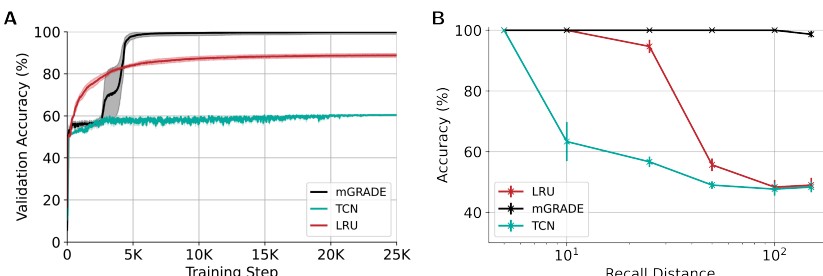

Figure A5: **mGRADE solves Flip-Flop modeling task better than TCNs and non-gated RNNs. A)** Validation accuracy (mean ± stde) over training steps. **B)** Recall accuracy (mean ± stde) for different recall distances.

### B.4.3 RECALL DISTANCE

In addition to the results reported in Section 3.2, we evaluated how well the model recalls the most recent value after a `w` given increasing distances between the `w` and `r` (the *recall distance*). For this, we construct Flip-Flop strings with one `w` at the beginning and a `r` in the middle, with different numbers of `i` symbols (with corresponding value symbols) in between. Fig. A5B shows the recall

accuracy of each of the models trained using the setup described above and tested on different recall distances. Note that the only model consistently performing accurate recall over distances of up to 100 timesteps is mGRADE. Even the LRU model decreases in accuracy with increasing recall distance, demonstrating the utility of a gated recurrent component.

## B.5 SELECTIVE COPYING TASK

### B.5.1 TRAINING AND EVALUATION DETAILS

**Task Description**  Following the training setup in Gu & Dao (2024) and Feng et al. (2025), we randomly generate sequences of 4096 timesteps at each training step. We train over $300,000$ steps. Each sequence contains 16 randomly distributed value symbols selected from an alphabet of size $n = 16$. After seeing the m symbol at the end of the sequence, the goal is to output the value symbols in the order they were received. For testing, we generated new sequences of the same length.

**Hyperparameters**  Following the architecture used in Gu & Dao (2024), we use a 2-layer mGRADE, comparing it to a 2-layer LRU augmented by DCLS and a 6-layer TCN (with an exponentially increasing receptive field). Just like the Flip-Flop modeling task, we use (Zucchet, 2024) as the basis for a drop-in LRU replacement into the mGRADE architecture. All models use encoders and decoders at the input and output, respectively. For the optimization, we use AdamW (Loshchilov & Hutter, 2019) for the weights, and Adam (Kingma & Ba, 2017) for the biases, the normalization layers, and the positions of the DCLS. We also use a cosine annealing learning rate scheduler without warmup. For the **z** gate biases, we use the Uniform Gate Initialization (UGI) initialization from Gu et al. (2020), while using the traditional zero initialization for all other biases. We initialize the weights with a truncated normal distribution, with a standard deviation set to $\sqrt{1/\text{fan\_in}}$. Other hyperparameters are outlined in Table A4. Reported results are averaged over 2 seeds (because of the compute-intensive nature of this task).

Table A4: Hyperparameters for the Selective Copying Task. L: number of layers. D/H: model dimensionality and hidden state size per layer (no hidden state expansion for any of these models). K: kernel count. $\Gamma$: kernel length (per layer from input to output for the TCN). LR: learning rate. B: batch size. Training Steps: number of batches presented during training. WD: weight decay.

| Parameter | L | D/H | K | $\Gamma$ | LR | B | Training Steps | WD |
|---|---|---|---|---|---|---|---|---|
| mGRADE | 2 | 64 | 32 | 128 | 0.001 | 64 | 300,000 | 0.1 |
| LRU + DCLS | 2 | 64 | 32 | 128 | 0.001 | 64 | 300,000 | 0.1 |
| TCN | 6 | 100 | 128 | 128/256/512/1024/2048/4096 | 0.003 | 64 | 300.000 | 0.0 |

**Loss Metrics**  Cross-entropy loss is used over the final outputs after the m symbols. The final accuracy is evaluated on randomly generated test sequences of the same length.

**Results**  Fig. A6 shows the accuracy over training steps for each model.

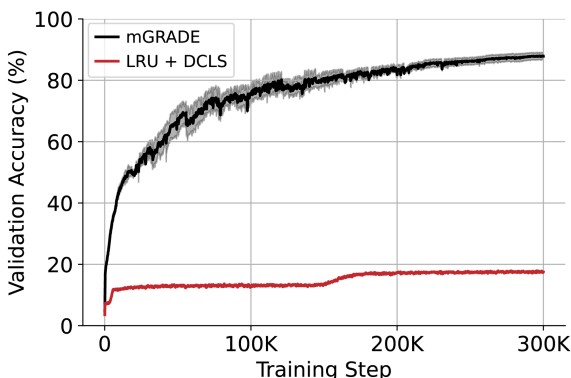

Figure A6: **mGRADE solves the Selective Copying Task better than LRU.** Validation accuracy (mean ± stde) over training steps.

## C  LRA AND GSC SETUP

### C.1  HYPERPARAMETERS

In Table A5, we provide the hyperparameters used for the mGRADE results on LRA and GSC reported in Table 3 and Table 4. We use the encoder and decoder at the input and output, respectively, as well as an MLP and layer normalization in each mGRADE layer. For the optimization, we use AdamW (Loshchilov & Hutter, 2019) for the weights, and Adam (Kingma & Ba, 2017) for the biases, normalization layers, and DCLS positions. In addition, the learning rate was scaled by 5 for the DCLS positions. For all tasks, we use a cosine annealing with linear warmup learning rate scheduler. We use two initialization schemes for the **z** gate biases, the traditional zero initialization and the UGI from Gu et al. (2020). We selected the zero-initialization for all the other biases. We use a truncated normal distribution for all weights (except the ones of the temporal convolution block), with a standard deviation set to $\sqrt{1/\text{fan\_in}}$. For the temporal convolution block, we set $\sqrt{\alpha/K}$ where $\alpha$ is a scaling hyperparameter. We use gradient clipping for every task except for Pathfinder, with a threshold of 10. For Pathfinder we use gradient global normalization with a threshold at 2. We did not use dropout. Finally, for the Image task, we introduced an extra linear layer at the output of the minGRU component and before the addition with the corresponding skip connection.

In Table A6, we provide the hyperparameters used for the results of our implementation of a fully causal and parameter-optimized LRU on LRA reported in Table 3. We use the code supplied in (Zucchet, 2024) without modifications, except for the Retrieval task which requires a specialized decoder. For this task, we use the specialized decoder for the Retrieval task from the official repository for (Smith et al., 2023) as a drop-in replacement for the default decoder used in (Zucchet, 2024). As in Orvieto et al. (2023), we use a cosine annealing with linear warmup learning rate scheduler as well as AdamW (Loshchilov & Hutter, 2019) for weight optimization and Adam (Kingma & Ba, 2017) for biases. For all internal recurrent parameters, a smaller learning rate was used, determined by a factor $\beta$ of the original learning rate. We also used the ring initialization from Orvieto et al. (2023), ensuring that the transition matrix eigenvalues were initialized between $r_{min}$ and $r_{max}$. For all these LRU-specific hyperparameters, we used those reported in the Orvieto et al. (2023), modifying only the hidden and model dimensionality to match our mGRADE implementation in terms of memory footprint. The learning rates were optimized using a base 10 logarithmic grid search. For our causal LRU, we only reported the highest validation accuracy (57.4%) reached given that our implementation was not able to significantly exceed chance level (50%) in terms of test accuracy, despite sweeping the hidden and model dimensionalities (reaching model sizes of up to 2M), the ring initialization (using $r_{min} = 0.9$ and $r_{max} = 0.999$ as well as $r_{max} = 0.9999$), the base learning rates (from 0.1 to 0.00001), and training for more than 800 epochs. Note that the original LRU experiments only report results on Pathfinder using a bidirectional model (Orvieto et al., 2023).

Finally note that LRA also includes the PathX task, which we do not evaluate for either mGRADE or our LRU implementation. The models reported in Table 6 that solve PathfinderX all use (acausal)

bidirectional processing. Since mGRADE is designed for causal, streamed processing, this experiment was outside of our scope. Nevertheless, mGRADE's promising results on the GSC dataset (which uses a sequence length that is as long as that of PathX, 16K) hint at mGRADE's potential suitability even to this task.

Table A5: Hyperparameters used for the reported mGRADE results on LRA. L: number of layers. D/H: model dimensionality and hidden state size per layer (no hidden state expansion for any of these models). K: kernel count. $\Gamma$: kernel length. LR: learning rate. B: batch size. Epochs: max epochs set for the run. WD: weight decay. ZBI: **z** gate bias initialization. $\alpha$: scaling factor for DCLS weight initialization. WU: number of epochs for the learning rate linear warmup.

| Parameter | L | D/H | K | $\Gamma$ | LR | B | Epochs | WD | ZBI | $\alpha$ | WU |
|---|---|---|---|---|---|---|---|---|---|---|---|
| ListOps | 6 | 32 | 2 | 16 | 0.003 | 64 | 100 | 0.1 | UGI | 0.05 | 10 |
| Text | 6 | 32 | 2 | 8 | 0.002 | 32 | 100 | 0.1 | zero | 0.25 | 10 |
| Retrieval | 3 | 64 | 2 | 8 | 0.003 | 32 | 20 | 0.1 | UGI | 0.05 | 4 |
| Image | 6 | 128 | 8 | 256 | 0.004 | 64 | 100 | 0.1 | zero | 0.1 | 10 |
| Pathfinder | 6 | 128 | 8 | 256 | 0.003 | 64 | 100 | 0.02 | zero | 1 | 10 |
| GSC | 6 | 64 | 16 | 64 | 0.0005 | 8 | 40 | 0.05 | UGI | 0.25 | 4 |

Table A6: Hyperparameters used for the results of our LRU reproduction on LRA. L: number of layers. D/H: model dimensionality and hidden state size per layer. $r_{min}/r_{max}$: minimum and maximum eigenvalue initialization radii. LR: learning rate. $\beta$: scaling factor for transition matrix learning rate. B: batch size. Epochs: max epochs set for the run. WD: weight decay. WU: number of epochs for the learning rate linear warmup.

| Parameter | L | D/H | $r_{min}/r_{max}$ | LR | $\beta$ | B | Epochs | WD | WU |
|---|---|---|---|---|---|---|---|---|---|
| ListOps | 6 | 33/32 | 0.0/0.99 | 0.001 | 0.5 | 32 | 100 | 0.05 | 10 |
| Text | 6 | 34/32 | 0.5/0.9 | 0.001 | 0.1 | 32 | 100 | 0.05 | 10 |
| Retrieval | 6 | 48/64 | 0.5/0.9 | 0.001 | 0.5 | 64 | 25 | 0.05 | 4 |
| Image | 6 | 160/256 | 0.9/0.999 | 0.01 | 0.25 | 64 | 100 | 0.1 | 10 |

## C.2 ACTIVITY BUFFER MEMORY FOOTPRINT

In this section, we explain how we compute the total buffer memory used by the baseline models in Table 3 and Table A7. S4 (Gu et al., 2022b), DSS variants (Gupta et al., 2022), Liquid-S4 (Hasani et al., 2023) all implement a similar architecture where H single-input, single-output SSM heads of size N are used in parallel. Thus, the amount of memory used for all recurrent hidden states is given by the formula $L \times (H \times N)$. S4-LegS (Gu et al., 2022b) uses H bi-directional SSM heads of size N in parallel. Thus, the amount of memory used by all states is given by the formula $L \times (H \times 2N)$. S5 (Smith et al., 2023), LRU (Orvieto et al., 2023) use only a single head multi-input, multi-output SSM of size N. Thus, the amount of memory used by all states is given by the formula $L \times N$. HGRN (Qin et al., 2023) uses a similar architecture to mGRADE's gated recurrent block, extended with complex states. Thus, the amount of memory used by all states is given by the formula $L \times (2H)$. The long-convolution models, SGConv (Li et al., 2023) and MRConv (Cunningham et al., 2024), need to buffer the activity of each neuron at each timestep, thus the amount of memory used by all states is given by the formula $L \times H \times T$ (where $T$ is the sequence length).

## C.3 PARAMETER COUNT FOR REAL-TIME PROCESSING ON EDGE DEVICES

In this section, we explain how real-time processing on edge devices impose taking into consideration some aspects that do not apply when running inference on GPUs. When processing inputs in real-time, it is too memory-expensive to save the entire sequence before processing it, let

alone buffering the activities of each neuron for the entire sequence length. For this reason, besides mGRADE, only S4 (Gu et al., 2022b), DSS (Gupta et al., 2022), Liquid-S4 (Hasani et al., 2023), S5 (Smith et al., 2023), and LRU (Orvieto et al., 2023) could be deployed on an embedded device and we report the adjusted numbers for these architectures in Table A7. We leave out bi-directional architectures as we care for causal processing. We also leave out the convolution architectures too as Table 3 shows that their activity buffers memory footprint is already above 1M for all tasks.

In S4 (Gu et al., 2022b) and Liquid-S4 (Hasani et al., 2023), the recurrent matrix is parametrized as a Diagonal Plus Low Rank (DPLR) matrix $A = \Lambda - PP^*$. This means that it is parametrized by two vectors of dimension $N$ (state dimension). However, when running in step-by-step recurrent mode on an embedded device, A would need to be instantiated into a full $N \times N$ matrix, which increases the number of effective parameters substantially. For example on the Image task, the number of parameters of S4 (Gu et al., 2022b) increases from 3.4M to 15.6M (i.e a factor of 4.5). Similarly to deploy mGRADE on an embedded device, we would need to fully materialize the DCLS kernels into vectors of dimension $\Gamma$. This increases the number of parameters of mGRADE from 712K to 896K on the Image task (i.e, a small 25% increase). With these results, we confirm that mGRADE is the architecture with the smallest memory footprint.

Table A7: **Fully Instantiated Memory Footprint for Recurrent Embedded Deployment.** Compare to Table 3. Note that S5 and HGRN use bidirectional input processing.

| Model | ListOps Params. / Buff. Act. | Text Params. / Buff. Act. | Retrieval Params. / Buff. Act. | Image Params. / Buff. Act. | Pathfinder Params. / Buff. Act. |
|---|---|---|---|---|---|
| S4 (Gu et al., 2022b) | 3.3M / 49K | 1.2M / 16K | 7.3M / 98K | 15.6M / 197K | 7M / 98K |
| DSS$_{\text{SOFTMAX}}$ (Gupta et al., 2022) | 206K / 49K | 152K / 16K | 888K / 98K | 2.0M / 197K | 601K / 98K |
| DSS$_{\text{EXP}}$ | 206K / 49K | 152K / 16K | 888K / 98K | 2.0M / 197K | 601K / 98K |
| DSS$_{\text{EXP-NO-SCALE}}$ | 206K / 49K | 152K / 16K | 888K / 98K | 2.0M / 197K | 601K / 98K |
| Liquid-S4 (Hasani et al., 2023) | 373K / 8K | 182K / 4K | 7.6M / 98K | 813M / 1.6M | 7.3M / 98K |
| S5 (Smith et al., 2023) | 190K / 1.5K | 1.3M / 1.1K | 772K / 1.5K | 5.1M / 2.3K | 1.1M / 1.5K |
| LRU (Orvieto et al., 2023) | 190K / 1.5K | 1.3M / 1.1K | 772K / 1.5K | − / − | 1.1M / 1.5K |
| HGRN (Qin et al., 2023) | 84K / 0.4K | 878K / 1.0K | 115K / 0.3K | 20.6M / 6.1K | 1.3M / 1.5K |
| mGRADE | 42K / 3K | 45K / 1.5K | 105K / 1.7K | 896K / 197K | 796K / 197K |

# D    EVALUATION ON NEUROMORPHIC SPIKING DATASET

To rigorously evaluate mGRADE's claims of parameter and memory efficiency in strictly low-resource environments, we extended our benchmarking to the neuromorphic domain. While Spiking Neural Networks (SNNs) and specialized event-based architectures are often cited as the standard for low-power temporal processing, we hypothesized that mGRADE's hybrid approach of combining lightweight gated recurrence with learnable delay convolutions could offer superior efficiency without requiring the specialized hardware or complex training dynamics often associated with SNNs. We selected the Spiking Heidelberg Digits (SHD) dataset, a standard neuromorphic benchmark consisting of approximately 10k recordings of spoken digits (0–9) in both English and German, resulting in 20 classes (Cramer et al., 2020). The dataset is constructed using an artificial cochlear model which results in 700 input channels. SHD is highly relevant for this evaluation because it requires the detection of precise temporal patterns within sparse spike trains to achieve good classification accuracy.

## D.1    EXPERIMENTAL SETUP

We follow the same data preprocessing pipeline established in the DCLS-delays work (Hammouamri et al., 2024), employing spatio-temporal binning to reduce input dimensionality. Input neurons were reduced from 700 to 140 by spatially binning over every 5 neurons. For the temporal dimension, we utilized zero right-padding to ensure a fixed window length for all sequences in a batch. Crucially, regarding the temporal resolution, the original DCLS-delays work reported using a discrete time-step of $\Delta t = 10$ ms. However, when using their official implementation with this reported configuration, we were unable to reproduce their state-of-the-art results. Through our own ablation of the baseline's configuration, we discovered that a finer temporal resolution of $\Delta t = 5$ ms was necessary to achieve performance parity with their reported figures. To ensure a fair and direct comparison, we therefore adopted this empirically verified configuration ($\Delta t = 5$ ms) for both the baseline reproduction and our mGRADE model. Consequently, to maintain the equivalent temporal receptive field (250ms), the sequence length (buffer size) was increased to 50 steps. We trained an mGRADE model using

the aforementioned preprocessing (with 5 ms bins) with two recurrent layers of 64 hidden units each, operating over a sequence length (buffer size) of 50 steps. Crucially, while the DCLS-delays baseline relies on a single delay element per synapse (1 kernel count), our mGRADE configuration utilizes 10 learnable delay taps within the convolution window. This design allows mGRADE to sample the input history more densely while maintaining a compact recurrent state. We applied a dropout rate of 10% for regularization. Finally, the output predictions are generated via a summation of the logit outputs across the entire sequence length, ensuring reproducibility with the DCLS-delays model. All reported results for both mGRADE and the reproduced DCLS-delays baseline are averaged over 5 random seeds.

Table A8: **Classification Accuracy and Parameter Efficiency on the Spiking Heidelberg Digits (SHD) Dataset.** We compare mGRADE against state-of-the-art efficient spiking architectures.

| Model | Params. | Buffer Size | Test Accuracy |
|---|---|---|---|
| SpikCommander (Wang et al., 2025) (1L-8-128) | 190K | - | **96.41%** |
| EventSSM (Schöne et al., 2024) (6L-64) | 400K | - | 95.90% |
| DCLS-delays (Hammouamri et al., 2024) (2L-256) | 200K | 50 | 93.95% $\pm$ 0.72 |
| mGRADE (2L-64) | **64.4K** | 50 | 93.77% $\pm$ 0.23 |

## D.2 RESULTS

The results, summarized in Table A8, demonstrate that mGRADE is highly competitive with specialized spiking architectures while being significantly more parameter-efficient. mGRADE achieves an accuracy of 93.77%, which is statistically comparable to the reproduced DCLS-delays model (93.95%), yet it does so with 3x fewer parameters (64.4K vs. 200K). Furthermore, compared to other high-performing models like EventSSM (400K params), mGRADE requires roughly 6x fewer parameters to achieve competitive performance. This evaluation successfully demonstrates that mGRADE establishes a new standard for parameter efficiency among this group while delivering competitive accuracy. It confirms that the combination of learnable delay embeddings (DCLS) and minimal gating (minGRU) is a powerful and highly efficient approach for sequence modeling in memory-constrained environments.

## E   mGRADE ANALYSIS

### E.1   ABLATION STUDY

In Section 3, we formally motivated the need for the temporal convolution and the gated recurrent component of mGRADE to tackle long-range dependency tasks. Table A9 compares the performance of mGRADE to architectures using only recurrent or convolutional components (using a mean pooling operator). Note that all architectures use the best hyperparameters found for mGRADE, scaling the hidden dimensionality to match parameter size. This means that kernel lengths and kernel counts were matched across mGRADE's convolution component and the purely convolution-based architectures for each task. The pure convolution-based models are the TCN, consisting of stacked causal dilated temporal convolution layers, and the DCLS model, made up of stacked causal temporal DCLS layers, which can be thought of as mGRADE without the recurrent component. For the pure gated recurrent architecture, we simply remove the convolutional component from the mGRADE layers, leaving us with the minGRU (Feng et al., 2025). We focus on the ListOps, Image, and Pathfinder tasks from LRA as (Orvieto et al., 2023) already showed that pure linear RNNs could solve the Text and Retrieval tasks to around 89%. Both the TCN and the DCLS models achieve good performance on Image while falling short on ListOps. On the other hand, minGRU achieves a better result than mGRADE on ListOps, while performing poorly on the Image task. Besides mGRADE, none of these architectures learn on Pathfinder. Our results are consistent with prior work demonstrating that purely convolution-based models (with around 10M parameters) achieve up to 83.62% on Image but only up to 72.28% and 52.93% on Pathfinder and ListOps, respectively (Miralles-González et al., 2025). Overall, our ablation study validates the the-

oretical motivations for each component of an mGRADE layer and showcases the synergy between the temporal convolution and gated recurrent components.

Table A9: **Ablation of mGRADE's component on the LRA benchmark.** The total memory footprint ("Total") at inference time is differentiated from the parameter count ("Params") as explained in Section 2. All architectures use the best hyperparameters found for mGRADE, we only scaled the layer width H to match the number of parameters.

| Model | ListOps | | Image | | Pathfinder | |
|---|---|---|---|---|---|---|
| | Acc | Params. / Buff. Act. | Acc | Params. / Buff. Act. | Acc | Params. / Buff. Act. |
| TCN | 39.6 | 45K / 4K | 85.3 | 727K / 220K | × | 867K / 230K |
| DCLS | 43.8 | 41K / 3K | 86.2 | 526K / 220K | × | 525K / 240K |
| minGRU | **62.5** | 40K / 192 | 66.0 | 697K / 786 | × | 597K / 786 |
| mGRADE | 61.9 | 40K / 3K | **87.1** | 712K / 197K | **94.9** | 612K / 197K |

### E.2 LEARNED KERNELS

We analyze the learned positions in the DCLS kernel across layers to understand how mGRADE adapts its temporal convolution mechanism to different task structures. Figure A7 shows the distribution of learned delay positions across the kernel (x-axis) and across hidden channels (y-axis) for models trained on the LRA Image, Pathfinder, and ListOps tasks (as done in Appendix E.1).

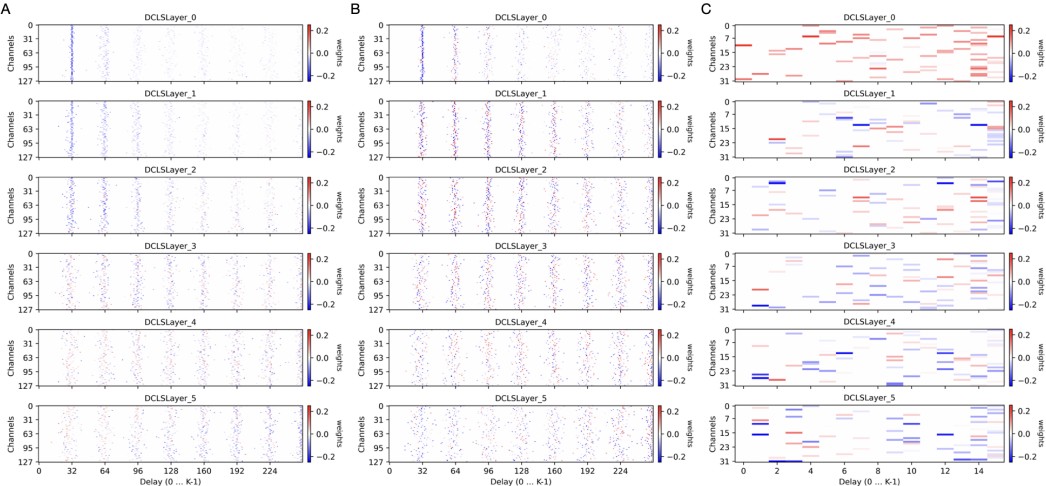

Figure A7: **Learned DCLS delay positions across layers for LRA benchmark.** Each panel shows the distribution of learned delay positions (x-axis) across hidden channels (y-axis) for all layers (0-5, top to bottom) of models trained on (A) Image, (B) PathFinder, and (C) ListOps. Color intensity indicates weight magnitude (red: positive, blue: negative). **A)** Image (kernel size 256, 8 elements, 128 hidden dims): Positions cluster around vertical bands at positions 32, 64, and 96, which for a $32 \times 32$ image correspond to the pixels in the same column but one, two, and three rows above respectively, demonstrating learned preference for local structure. Deeper layers show increasing dispersion around the vertical bands (offsets ranging from 0-10) while maintaining locality. **B)** Pathfinder (kernel size 256, 8 elements, 128 hidden dims): Delays disperse much more across the full 0-256 range, reflecting the sparse, non-local nature of path detection where relevant features (blobs and connecting paths) appear at arbitrary locations. Moderate vertical banding suggests some channels specialize for identifying the blobs at early layers. **C)** ListOps (kernel size 16, 2 elements, 32 hidden dims): Full utilization of the 0-15 delay range across layers. The sparse sampling (2 of 16 positions per channel) enforces efficient information aggregation for bracket matching and operator precedence.

### E.2.1 LRA Image: Emergence of Spatial Local Processing on Sequential Images

For the LRA Image task (Fig. A7A), the learned positions reveal a hierarchical local feature extraction pattern. In the early layers (0-2), positions sharply cluster around a delay of 32 timesteps. When processing the $32 \times 32$ images used in LRA sequentially, this delay precisely corresponds to the pixel directly above the current input pixel. This concentration on immediate spatial neighbors (whether above or below) resembles the local receptive fields of classic 2D Convolutional Neural Networks (CNN), suggesting that mGRADE, through training, automatically tends towards spatial locality for image processing despite the sequential presentation of the image. Deeper layers (3-5) maintain this locality bias, but with increased dispersion in time, effectively expanding the receptive field and with it the spatial context (while still remaining local). The distinct vertical bands observed across channels indicate specialized feature detectors, some channels consistently attend to immediate neighbors while others look slightly further with a few specific offsets. This hints towards the model using its learnable positions to capture irregularly spaced patterns and a larger range of spatial frequencies within the spatially local receptive field it builds over the image.

### E.2.2 LRA Pathfinder: Distributed Search for Sparse Structure

In contrast, the delays learned by the model trained on the LRA Pathfinder task (Fig. A7B) exhibit slightly different patterns that reflect the task's more sparse, less local structure. While still exhibiting some vertical clustering, delays appear more dispersed around the clusters compared to the Image task. Particularly deeper layers show significantly weaker clustering than the corresponding layers in the model trained on Image. This distribution suggests that the model cannot rely only on local patterns, as the relevant features (two dots and their connecting path in a $32 \times 32$ image) can appear at arbitrary pixel locations across samples. Early layers show moderate clustering at certain delay values visible as a few strong vertical bands, especially in layers 1 and 2. Later layers maintain broader delay coverage, suggesting they aggregate evidence across multiple spatial scales to determine path connectivity. The weaker locality bias in the positions indicates that spatially adjacent pixels in the Pathfinder images provide less predictive power than is the case for the natural images in Image.

### E.2.3 LRA Listops: Adaptation to Nested Dependencies

The learned positions for ListOps (Fig. A7C) reveal a sophisticated strategy for parsing symbolic structures, which stands in contrast to the patterns seen in the Image task. A key feature is how mGRADE's temporal receptive field spans the entire kernel length, all 16 timesteps across all layers. This reflects the task nature, where meaningful dependencies between operators, operands, and matching brackets occur at variable distances but remain bounded to a range of around 10 timesteps on average (Nangia & Bowman, 2018). In the initial layers (0-1), the model establishes a strong inductive bias by concentrating weights at diverse positions, combining more local information (delays close around 3) with information at the boundaries of the kernel (positions around 14). In contrast, the deeper layers exhibit a more distributed pattern, with channels dedicating kernel elements sparsely to specific and non-local delays. This sparse distributed sampling allows the model to track multiple long- and mid-range dependencies simultaneously and efficiently.

These contrasting solutions demonstrate mGRADE's ability to adapt its information aggregation mechanisms to task structure without overpowering inductive biases (given that the kernel length is large enough). For structured data with strong spatially local correlations (Image), the model converges to classic 2D CNN-like local processing. For tasks requiring global reasoning over sparse features (Pathfinder), it maintains broader temporal coverage. Listops, on the contrary, requires a hybrid of mid-range and long-range aggregation in deeper layers. This adaptive behavior arises from enabling the delays to be learnable, validating DCLS as a flexible alternative to TCNs with fixed dilation rates, and further justifying our decision to use it in mGRADE.

