# OpenReview forum: "mGRADE: Minimal Recurrent Gating Meets Delay Convolutions for Lightweight Sequence Modeling"
_ICLR.cc/2026/Conference — Submitted to ICLR 2026_

### Official Review · Reviewer_BrFx · 2025-10-31

**Soundness:** 2
**Presentation:** 2
**Contribution:** 2
**Rating:** 2
**Confidence:** 4

**Summary:**

This paper presents mGRADE, an examination of a hybrid architecture combining recurrent models with convolutions.  The objective is to use the convolutions as a short-term processing mechanism, and then use a lighter recurrent network to process long-range dependencies.  The core mGRADE layer is a 1-d convolution (in time), a minGRU recurrent layer (providing gating), and a position-wise FFN.  A stacked architecture is then built out of this.  Some simple theoretical exploration of fundamental tasks is presented so explore the core properties.  The model is then benchmarked on LRA and achieves very competitive results despite using fewer parameters, smaller memory footprints and being strictly unidirectional.

**Strengths:**

1. Reducing (or even remarking on) memory consumption and memory complexity is often underserved in the SSM literature.  A lot of SSM architectures eulogize constant time-complexity per step, but assume access to massive computational hardware.  This is a nice examination of that angle.
2. I like how quickly the paper gets into the meat of the topic.
    (I will note, mind you, it could use another paragraph of signposting on what you will show and why:  some combination of a concrete problem specification and notation section;  a list of key elements to prove or introduce;  expanded discussion of the shortcomings in other methods you will overcome;  etc.  Just to prime the reader for what is coming in a bit more detail.  It is a bit of a "smash-cut" into technical detail.)
3. The banner figure is a good for introducing the architecture.
4. There are a good number of experiments and ablations presented.

**Weaknesses:**

# Summary
I find this paper hard to place.  Memory and edge applications are super important to consider, and _are_ underserved in the SSM literature.  However, this paper finds itself in an awkward in-between space where: (1) it is comparing to SSMs not optimized for memory consumption; and (2) doesn't compare to memory-efficient architectures; and (3) doesn't actually evaluate the edge suitability of any of the models.  Instead of being in the overlapping section of the three circles in a Venn diagram, it is in a region where no regions overlap.

I therefore struggle to convince myself that this paper fully and accurately encapsulates the current state of the field(s).  Ultimately, the paper (as presented) states:  "here is _a_ more memory efficient SSM," without fully fleshing out the bounds or expectations on practical performance.  Therefore, I unfortunately cannot endorse this paper for publication in its _current_ form.

With that said, I think the core of a _very_ good paper is here, and I strongly encourage the authors to continue working on it, expanding the tie-ins and evaluation, because it could be a very compelling study.  Good luck.

# Major Weaknesses
1. **Comparing with performance-optimized SSMs**:  A lot of the results quoted didn't consider parameter complexity (in fact, most of the architectures were developed with roughly matching parameter complexity).  It is therefore a bit of an unfair comparison to claim superiority in terms of parameter complexity, when it is not clear how well, e.g., LRU would do with 40k parameters.  It is impressive that you get so close with 20% of the parameters, but the comparison is, in my opinion, incomplete.  Equally, although this isn't part of the initial pitch of your paper, I would be interested to know how parameter-matched mGRADE compares in terms of performance.  To answer this, I'd need to see numerous experiments sweeping across parameter counts, across different hyper parameter and architectural configurations.
2. **Absence of efficient architectures**:  I am not an expert in highly compressed neural architectures, but I know there are things like MobileNet, neuromorphic computation, spiking neural networks, low-rank factorized models etc that are specifically designed for low-resource environments.  I therefore don't know if I'm really convinced by the breadth of the evaluation of your claims of parameter and memory efficiency.  Yes, you have made a possibly more parameter-efficient SSM, but I feel that this is a very narrowly scoped claim that maybe doesn't rise to the level of significance required for ICLR.
3. **Absence of edge-evaluation**: Despite discussing efficient computation throughout, none of the architectures are actually deployed on edge hardware.  I'd _love_ to see an evaluation of these architectures, for instance, running on a Raspberry Pi, and recording maximum sequence length, latency, failure rate etc.  Simply claiming improved memory efficiency, while executing all experiments on a 24GB GPU feels like missing an opportunity.  If you could show your network deployed on these architectures where others can't, then that would be huge strengthener to your claims.
4. Absence of Path-X:  I respect the authors may not have access to huge computational hardware, but unfortunately Path-X is where differences in architectures is really observed:  many modern architectures/hyperparameters that perform comparably on the other LRA tasks fail critically on Path-X.
5. Long convolutions?:  I notice that the convolutions in Image and Pathfinder are very long -- specifically 1/4 of the input length.  I query therefore how much of the lifting this convolution is doing, and how much the recurrence is even needed.  I would like to see a baseline of how a purely-convolutional model (using the $\Gamma$ you use) with a suitable pooling operator (mean/max) performs.

# Minor Weaknesses/Typographical Comments
1. The minimal gating is just a direct application of minGRU, this should be advertised more clearly and earlier.  I don't love the way some of the "minimal gating" is advertised earlier, making it feel like it is a novel contribution.
2. Table A6 is very important, I think.  It shows that the actualized gains are far lower than advertised (if at all).  This table should be brought up to the main paper.
3. Why are HGRN and S5 omitted from Table A6?
4. Mamba is also missing from all your evaluations, why?
5. Are you able to run all the baselines in Table 2?  It is a bit of a shame to have missing entries.

# Missing Literature
1. MEGA (and MEGA-Chunk) [Ma+, 2023] should be (at least qualitatively) compared to, as they use a lightweight 1-D convolution layer before inputting into a chunked transformer.  This would achieve a similar effect.

**Questions:**

I invite the authors to respond to my comments in "weaknesses".

Q1. There is a known equivalence between convolutions and SSMs (this is the basis of S4!).  Is it just that directly parameterizing the convolutions is more parameter efficient?  Why can't you achieve this fully recurrent?  I think at least qualitative discussion of this is absent.

---

> ### Author Response · Authors · 2025-11-20
> **Reply - Part 1**
>
> We thank the reviewer for their careful assessment of our submission and for the constructive feedback. After reviewing the comments, we realized that we may not have sufficiently explained why we believe that our paper represents a significant contribution to the field of efficient sequence models beyond merely presenting a more memory efficient SSM. Accordingly, we would like to highlight two central contributions of our work, which underlie mGRADE’s ability to achieve high performance with remarkable memory efficiency.
>
> (i) the temporal convolution in mGRADE incorporates learnable spacings, which is fundamentally different from the standard short 1D-convolutions used in many sequence model backbones. This mechanism enables the model to implement a flexible, learnable delay embedding, connecting directly to classical dynamical systems theory (e.g., Takens’ theorem) and providing formal guarantees about its ability to capture short-range temporal structure. In contrast to the dense 1D-convolutions typically used in SSM backbones, our learnable-spacing convolution directly modulates the temporal positions of inputs. In particular, it transforms the classical convolution operation into a dynamic delay operator that directly learns the appropriate temporal geometry of the data by moving kernel elements in time rather than only modifying weights at the respective kernel positions. In addition to aiding learning by providing an inductive bias for temporal structure in the data, this training method for the temporal convolution is also more parameter efficient by employing the temporal positions as parameters rather than the full sets of weights in dense kernels.
>
> (ii) In combination with the minimal gated recurrent block, mGRADE explicitly separates short-term (delay embedding) and long-term (gated recurrence) dynamics, promoting functional specialization across modules and yielding a model whose internal computation is more interpretable than standard sequence models.
>
> These aspects, i.e., the novelty of the learnable-spacing temporal convolution and the resulting interpretability benefits, were not commented on by the reviewer, and we acknowledge that we did not emphasize them clearly enough in the original submission. We have strengthened this discussion in the revised version, and in this light, we address your concerns in detail below
>
> > Major Weakness 1: Comparing with performance-optimized SSMs [...] e.g., LRU with 40k parameters...
>
> Thank you for this comment. While early SSM models like S4 were motivated by minimizing the memory complexity, you rightfully note that parameter complexity was not a central consideration (particularly with respect to results obtained on the LRA benchmark). As such we believe your suggestion to compare the performance of SotA SSM architectures optimized for parameter count with mGRADE would greatly strengthen our claims regarding parameter efficiency. Following your suggestion, we will optimize a parameter-matched LRU on each of the LRA tasks, reporting the best achieved accuracy as a fair parameter-matched comparison to mGRADE. We will update the manuscript and inform you as soon as the results are available.
>
> We would love to perform more comprehensive sweeps, including investigating the performance of mGRADE as it scales up, but due to our computational constraints we decided to prioritize mGRADE’s performance on time series benchmarks (since it aligns better with the edge processing purpose). As of now, we rely on the theoretical arguments, particularly regarding mGRADE’s selectivity which aligns with prior findings on Transformers and larger SSMs, as an indication that mGRADE’s performance would improve as it scales up.

---

> ### Author Response · Authors · 2025-11-20
> **Reply - Part 2**
>
> > Major Weakness 2: Absence of efficient architectures...
>
> Unfortunately, very little prior work evaluates models optimized for parameter efficiency on long range dependencies such as those in the LRA task. While parameter-efficiency may have not been the primary goal for the SSMs we compare to in Table 6, it has been used as an additional selling point (see for example Section 4.1 in the S4 paper [1] or Section 5 in [2]). As such, the performance-optimized SSMs shown in Table 3 are also the most parameter-efficient models able to handle LRA we could find in the literature. This explains the heightened interest in SSMs within the sequence modeling ASIC community (see for example, [3, 4, 5]).
>
> However, we agree that a comparison to architectures optimized for memory efficiency would greatly strengthen our claims. While MobileNet is optimized for efficient edge applications, it is primarily designed for image processing, not sequential tasks. Given that the pure TCN baselines we test in Table A.7 (and for many of the synthetic tasks) already underperform mGRADE in terms of accuracy, a sequence-adapted MobileNet is unlikely to fare significantly better. Thus, we believe that spiking neural networks (SNNs) are the most appropriate comparison since many of them are also designed for efficient sequence modeling. We are currently working on obtaining results comparing mGRADE to SotA SNNs and will update the paper with results once they arrive.
>
> > Major Weakness 3: ... If you could show your network deployed on these [edge hardware architectures] where others can't, then that would be huge strengthener...
>
> We agree of course that the next logical step for a novel architecture designed for memory efficient embedded systems is the actual deployment of the model on such devices. Our focus here lies on ASICs specialized for ultra-low-power sensory processing, for example, the ones indicated in the table below. These systems achieve their level of energy efficiency by being strongly tailored to specific model architectures, often sacrificing configurability for efficiency.  Designing and deploying on such ASICS is beyond the scope of this paper which focuses on the novel architecture and its theoretical underpinnings. We see this as a hardware-informed algorithm which is the first step in the journey of developing energy efficient and miniaturized ASICs for AI models.
>
> That being said, given the memory sizes of current ASICs available in the table below, we can see that (in contrast to many of the baseline models we compare ourselves to in Table 3 and A6) mGRADE's required memory footprint is in the same ballpark of what these types of systems can realistically accommodate (10s to 100s of thousands of parameters).
>
> If the reviewer thinks that a more detailed discussion of this topic and the required steps towards an ASIC implementation (e.g. model quantization and hardware-algorithm co-design) is of interest, we are happy to add a section to the appendix (and if the page limit permits to the discussion) based on a table like the one below.
>
> | | | | | | | | | |
> |:---:|:---:|:---:|:---:|:---:|:---:|:---:|:---:|:---:|
> | | [6] | [7] | [8] | [9] | [10] | [11] | [12] | [13] |
> | **Model Type** | LSTM+FFN | TCN | CNN | CNN | TCN+FFN | Spiking RNN | SNN | Delta-RNN |
> | **On-Chip Memory (for weights)** | 67kB | 132kB | 16kB | 349kB | 71kB | 138kB | 353kB | 48kB |
> | **\# Max. On-Chip Weights** | 32k | 400k | 32.8k | 2.1k | 133k | 132k | 194k | Depends on weight precision |
> | **Max. Weight Prec.** | 4bit | up to 8bit | 4bit | 16bit | 4bit | 8bit | 14bit | Configurable |
>
> *On-Chip Weight Memory refers to how many bytes are available for weights, routing, etc. (not including the activation buffer)*
>
> *# Max. On-Chip Weights refers to the actual maximum number of individual weight connections. This is separate from on-chip weight memory because many of these chips have hardwired network structures with fixed weight precision.*
>
> *Weight Precision refers to the highest possible weight precision.*

---

> ### Author Response · Authors · 2025-11-20
> **Reply - Part 3**
>
> > Major Weakness 4: Absence of Path-X...
>
> We thank the reviewer (as well as reviewer UQK5 in Q2) for pointing this out. We of course agree that having results on Path-X would strengthen our paper. However, we do not have access to a large amount of computational resources and therefore have to prioritize which tasks we will be tackling during this rebuttal process. We have assigned Path-X a lower priority compared to time series datasets for the following reasons:
>
> As far as we are aware there exist very few, if any, prior results of causal, unidirectional, streamed architectures that publish results on this tasks. All architectures in Table 6 (SotA for LRA) employ bidirectional inputs and therefore are not an appropriate comparison for our model (which is designed for causal and streamed input) on this task.
>
> The lack of prior results without bidirectionality hints to the fact that this task is extremely difficult for streaming and causal architectures.
>
> Runtimes of 36h for a single seed on our setups leads us to assign a low success chance for finding appropriate hyperparameters within a reasonable time for our model that would result in competitive performance on this task.
>
> > Major Weakness 5: ... how much of the lifting [is the] convolution doing, and how much the recurrence is even needed...
>
> In Table A7, we present an ablation study on mGRADE, presenting results on Image, Pathfinder, and ListOps from a purely recurrent minGRU and a pure convolution-based TCN and DCLS-parameterized TCN (with learnable spacings). The TCN and DCLS models use the same kernel lengths as mGRADE and mean pooling. We added a sentence in Appendix D.1 to clarify this. As can be seen, the TCN and DCLS models get fairly close to mGRADE’s performance on the Image task but still fall short by 1-2%. On Pathfinder, the convolution-based models fail to exceed chance-level, despite our best efforts. Prior work on solving Pathfinder and Image with pure 1-D convolution-based models required at least 10M parameters to reach 72.3% (Pathfinder) and 83.6% (Image) [14]. mGRADE can solve both tasks at higher accuracy with 2 orders of magnitude fewer parameters. We added a reference to these previous results in the text of Appendix D.1 as well.
>
> Note that Appendix D.2 contains a more comprehensive (and very interesting) discussion on how much and what kind of “lifting” the temporal convolution kernels are doing for Pathfinder, Image, and ListOps. Given our use of learnable spacings, the kernel structure adapts to the temporal structure of the task itself. For example, kernels that are local in 2D space emerge for the Image task (despite the image being presented as a 1D sequence) while for the ListOps task a temporal hierarchy across layers emerges matching the nested dependencies in ListOps.
>
> > Minor Weakness 1:  Minimal gating is just a direct application of minGRU, this should be advertised more clearly...
>
> We internally debated for a while what to call the gated recurrent component. Although the name “minGRU” was recently popularized by [15], this pattern of minimal gated recurrence has been proposed as early as [16] (as the strongly-typed RNN), used as part of the Quasi-RNN by [17], with [18] suggesting the use of parallel scan for efficient training (while calling it the gated impulse linear recurrent unit). As such we used “gated recurrence” and “minimal gating” to emphasize the general architectural pattern and its theoretical expressivity (and to avoid having to pick between the many alternative designations). However, based on your feedback we have updated the abstract, introduction, and model specification sections to highlight the fact that our “minimal gated recurrent component” is in fact equivalent to the minGRU and its predecessors.

---

> ### Author Response · Authors · 2025-11-20
> **Reply - Part 4**
>
> > Minor Weakness 2:  ...[Table A6] shows that the actualized gains are far lower than advertised [and] [...] should be brought up to the main paper.
>
> We thank the reviewer for their careful reading of Table A6 and for the suggestion to move it into the main paper. However, we are not fully certain how to interpret the comment that “the actualized gains are far lower than advertised (if at all)” in the context of this table. We suspect there may be a misunderstanding regarding what Table A6 is intended to demonstrate. The goal of Table A6 is to highlight a critical trade-off inherent in the parameter efficiency versus inference memory in many existing SotA SSMs. Existing SSMs achieve a low number of trainable parameters by adopting clever matrix parametrizations (e.g., the DPLR $A = \Lambda - PQ^*$ used in S4, which reduces $N \times N$ parameters to $3N$). However, for efficient, low-power real-time inference on edge platforms like FPGAs and ASICs, the weight-stationary computing paradigm [19] is essential, as data movement is orders of magnitude more costly than computation [20]. In this context, the full $N \times N$ recurrent matrix is statically instantiated in memory, leading to substantially higher instantiated parameter counts during deployment, despite the low trainable count. mGRADE must also instantiate additional parameters during deployment due to the parameterization of the learnable spacings which include a Gaussian spread over several timesteps. Crucially, the difference between the instantiated parameters and the trainable parameters is significantly smaller than the full $N \times N$ matrix instantiation required by discretized SotA SSMs. Among existing approaches, only DSS is comparably memory-efficient once instantiated, but it lags significantly in accuracy. Therefore, Table A6 further clarifies, arguably even more so than Table 3, that mGRADE addresses a critical gap by balancing competitive accuracy with a significantly reduced memory footprint for inference compared to other high-accuracy SSMs. If the reviewer prefers for Table 3 to report the number of instantiated parameters rather than the number of trainable parameters, we are happy to revise it accordingly; this change would, in fact, further strengthen our argument.
>
> > Minor Weakness 3: Why are HGRN and S5 omitted from Table A6?
>
> We appreciate the reviewer's suggestion to include HGRN and S5. We have revised Table A6 to include these architectures, which now clearly highlights a critical trade-off: causal constraints versus bidirectionality. HGRN and S5, while potentially achieving higher performance, rely on a bi-directional scheme that requires buffering the entire input signal. This design choice drastically increases the memory footprint and adds significant end-to-end latency, making them unsuitable for the real-time, streaming causal inference setting targeted by mGRADE. The revised Table A6 now serves as a comprehensive comparison that explicitly demonstrates how mGRADE maintains competitive accuracy while operating under the strict, low-latency requirements of a true causal system, unlike these bi-directional counterparts.
>
> > Minor Weakness 4: Mamba is also missing, why?
>
> While Mamba offers excellent performance on language modeling benchmarks, it was not necessarily designed to handle long range dependencies across the different modalities in the LRA dataset. According to the authors, it particularly struggles with handling long-range dependencies in images, performing comparably to S4 on the other LRA tasks [21]. [22] provides actual results using Mamba on selected LRA tasks (unfortunately without parameter numbers), showing good accuracy on Retrieval (90%) and Pathfinder (99%) but significantly lower performance than mGRADE on ListOps (33%) and Image (69%). Although this comparison is favourable to mGRADE in many respects, our focus lies on edge processing and not large language modeling, so we left Mamba out of Table 3. We added a sentence highlighting Mamba’s focus on language modeling in Section 5 (under “Linear Recurrent Models”).
>
> > Minor Weakness 5: Are you able to run all the baselines in Table 2?
>
> We have updated Table 2 to include the TCN baseline on the Selective Copying task. The H3 results were taken from the Selective Copying task in [23]. If time permits, we will try to run the public code of H3 on the Flip-Flop task.

---

> ### Author Response · Authors · 2025-11-20
> **Reply - Part 5 (+ references)**
>
> > Question 1. There is a known equivalence between convolutions and SSMs (this is the basis of S4!). Is it just that directly parameterizing the convolutions is more parameter efficient? Why can't you achieve this fully recurrent? I think at least qualitative discussion of this is absent.
>
> We agree with the reviewer that this topic warrants a short discussion in the paper. Accordingly, we have added a section to the appendix (see Appendix A.2) that we summarize in the following.
>
> It is true that temporal convolutions and SSMs are theoretically equivalent. However, during deployment on an embedded system, representing an arbitrary convolution with learnable spacings using a fully recurrent SSM is not as parameter efficient as using the convolution directly. To capture an arbitrary kernel of length $\Gamma$, a naively parameterized SSM would require a state size of $\Gamma$ and a corresponding $\Gamma \times \Gamma$ transition matrix, even if only $K \ll \Gamma$ weights are actually learnable. Thus, while it is theoretically possible to reformulate our convolution with learnable spacings as a fully recurrent SSM, it would decrease the memory efficiency. Note that this also relates to your questions about Table A6. In addition to the concerns about memory efficiency, reformulating the convolution component as an SSM would complicate the use of DCLS for learning the spacings which is a critical component of mGRADE’s overall architecture.
>
> ### Missing Literature
>
> We have added a short remark on MEGA to our “Related Works” in Section 5.
>
> ### References:
>
> [1] A. Gu et al. Efficiently modeling long sequences with structured state spaces. ICLR 2022.
>
> [2] J. Sieber et al. Understanding the differences in foundation models: Attention, state space models, and recurrent neural networks. Neurips 2024.
>
> [3] S. Siegel et al. IMSSA: Deploying modern state-space models on memristive in-memory compute hardware. ISCAS, 2025
>
> [4] S. Ward-Foxton. Applied Brain Research Demos first silicon for state-space models. In EE Times, 2025. https://www.eetimes.com/applied-brain-research-demos-first-silicon-for-state-space-models/
>
> [5] T. Lewis et al. Introducing tenn: Revolutionizing computing with energy efficiency. BrainChip, 2025. https://brainchip.com/introducing-tenn-revolutionizing-computing-with-an-energy-efficient-transformer-replacement/
>
> [6] J. S. P. Giraldo et al: A 65-nm speech-triggered wake-up soc for 10- µ w keyword spotting and speaker verification. In IEEE Journal of Solid-State Circuits 2020.
>
> [7] V. Jain et al: A tiny versatile system-on-chip with state-retentive mram for ml inference at the extreme edge. In IEEE Journal of Solid-State Circuits 2023.
>
> [8] F. Tan et al.  A 1.8% far, 2 ms decision latency, 1.73 nj/decision keywords-spotting (kws) chip incorporating transfer-computing speaker verification, hybrid-if-domain computing and scalable 5t-sram. In IEEE Journal of Solid-State Circuits 2025.
>
> [9] H. Yang et al. Fsl-hdnn: A 5.7 tops/w end-to-end few-shot learning classifier accelerator with feature extraction and hyperdimensional computing. ESSERC 2024
>
> [10] D. den Blanken and C. Frenkel. Chameleon: A MatMul-Free Temporal Convolutional Network Accelerator for End-to-End Few-Shot and Continual Learning from Sequential Data. In arXiv, 2025.
>
> [11] C. Frenkel and G. Indiveri. ReckOn: A 28nm Sub-mm2 Task-Agnostic Spiking Recurrent Neural Network Processor Enabling On-Chip Learning over Second-Long Timescales. ISSCC 2022
>
> [12] J. Park et al. A 65-nm 236.5 nJ/Classification Neuromorphic Processor with 7.5% Energy Overhead On-Chip Learning Using Direct Spike-Only Feedback. ISSCC 2019
>
> [13] S. Zhou et al. An 8.62-μW 75-dB DRSoC Fully Integrated SoC for Spoken Language Understanding. In IEEE Journal of Solid-State Circuits 2025
>
> [14] P. Miralles-Gonzalez et al. On the locality bias and results in the long range arena, arxiv 2025
>
> [15] L. Feng et al. Were RNNs all we needed?, 2025. URL https://openreview.net/forum?id=GrmFFxGnOR.
>
> [16] D. Balduzzi and M. Ghifary. Strongly-Typed recurrent neural networks. ICML 2016
>
> [17] J. Bradbury et al. Quasi-recurrent neural networks. ICML 2017
>
> [18] E. Martin and C. Cundy. Parallelizing linear recurrent neural nets over sequence length. ICLR 2018
>
> [19] A. Shafiee et al. ISAAC: A Convolutional Neural Network Accelerator with In-Situ Analog Arithmetic in Crossbars, ACM/IEEE 43rd Annual International Symposium on Computer Architecture (ISCA) 2016
>
> [20] M. Horowitz. Computing’s energy problem (and what we can do about it). ISSCC 2014
>
> [21] Albert Gu. Issue #282: Mamba in Long range arena (LRA). GitHub. 2024. https://github.com/state-spaces/mamba/issues/282
>
> [22] M. Beck et al. xLSTM: Extended Long Short-Term Memory. Neurips 2024
>
> [23] A. Gu and T. Dao. Mamba: Linear-time sequence modeling with selective state spaces. In First Conference on Language Modeling 2024

---

> ### Comment · Reviewer_BrFx · 2025-11-22
> **Staying Put**
>
> To the authors,
>
> Thank you for your detailed feedback.  Unfortunately, I am not swayed by any of the points you raise, or the many of the points in response to the other reviewers.
>
> The core of my critique remains: I do not think the paper is thorough enough in any one of the three directions identified for me to be happy endorsing that this paper should be committed to that line of work.
>
> I highly encourage the authors to keep working on this, however, because I think there is a valuable paper in here, it is just not there yet.
>
> Good luck,
>
> BrFx

---

> > ### Author Response · Authors · 2025-11-25
> > **Experimental Results Coming Up**
> >
> > We thank the reviewer for their follow-up and fully respect their decision not to change their score.
> >
> > We are currently running the experiments you had suggested with promising results and look forward to presenting them to you before Friday. We are highly motivated to continue improving this work by then, and if it is still not convincing, we would appreciate concrete feedback on which arguments and additional experiments would change the reviewer’s assessment.

---

> > > ### Author Response · Authors · 2025-12-03
> > > **Experimental Results - Tables**
> > >
> > > We provide the requested experimental results.
> > >
> > > We downscaled the LRU architecture for it to respect the same parameter count as mGRADE, and trained it on the LRA tasks in a causal way.
> > >
> > > | Model | ListOps Acc. | ListOps Params. / Buff. | Text Acc. | Text Params. / Buff. | Retrieval Acc. | Retrieval Params. / Buff. | Image Acc. | Image Params. / Buff. | Pathfinder Acc. | Pathfinder Params. / Buff. |
> > > |-------|--------------|--------------------------|-----------|------------------------|-----------------|----------------------------|------------|--------------------------|------------------|-----------------------------|
> > > | LRU (ours) | 58.3 | 42K / 0.2K | 85.9 | 46K / 0.2K | 86.8 | 118K / 0.4K | 84.3 | 1.2M / 2.3K | 57.4 | 1.1M / 1.5K |
> > > | mGRADE | 61.9 | 40K / 3K | 87.3 | 44K / 1.5K | 88.1 | 104K / 1.7K | 87.1 | 712K / 197K | 94.9 | 612K / 197K |
> > >
> > > To evaluate mGRADE against architectures tailored for edge efficiency, we additionally report results on the Spiking Heidelberg Digits dataset [1], a standard benchmark for spiking neural network models.
> > >
> > > | Model                                                         | Params.   | Buffer Size | Test Accuracy        |
> > > |---------------------------------------------------------------|-----------|-------------|-----------------------|
> > > | SpikCommander (1L-8-128) [2]                                  | 190K      | -           | **96.41%**           |
> > > | EventSSM (6L-64) [3]                                          | 400K      | -           | 95.90%               |
> > > | DCLS-delays (2L-256) [4]                                      | 200K      | 50          | 93.95% ± 0.72        |
> > > | **mGRADE (2L-64)**                                            | **64.4K** | 50          | 93.77% ± 0.23        |
> > >
> > > [1] B. Cramer et al. The Heidelberg Spiking Data Sets for the Systematic Evaluation of Spiking Neural Networks. IEEE Transactions on Neural Networks and Learning Systems, 2022.
> > >
> > > [2] J. Wang et al. SpikCommander: A High-performance Spiking Transformer with Multi-view Learning for Efficient Speech Command Recognition. AAAI, 2026.
> > >
> > > [3] M Schone et al. Scalable event-by-event processing of neuromorphic sensory signals with deep
> > > state-space models. ICONS, 2024.
> > >
> > > [4] I. Hammouamri et al. Learning delays in spiking neural networks using dilated convolutions with learnable spacings. ICLR, 2024
> > >
> > > We further report results on the raw-audio Google Speech Commands task, which matches the sequence length of Path-X and enables a direct comparison with long-range recurrent architectures.
> > >
> > > | Model                                       | Parameters / Buff.      | Causal            | Bidirectional        |
> > > |---------------------------------------------|--------------------------|-------------------|-----------------------|
> > > | S4-LegS [1,2]                                | 307K / 49K               | 93.6              | 96.1                  |
> > > | S4-FouT [1,2]                                | 307K / 49K               | 91.8              | 95.3                  |
> > > | S4D-LegS [1,2]                               | 306K / 49K               | 93.6              | 95.8                  |
> > > | S4D-Inv [1,2]                                | 306K / 49K               | 93.4              | 96.2                  |
> > > | S4D-Lin [1,2]                                | 306K / 49K               | 93.4              | _96.3_                |
> > > | Liquid-S4 [1,2]                              | _224K_ / _5K_            | **96.8**      | -                     |
> > > | S5 [1,2]                                     | 280K / **1K**            | -                 | **96.5**              |
> > > | **mGRADE**                                   | **198K** / 20K           | _94.7_            | -                     |
> > >
> > > [1] Uses complex numbers in the recurrence.
> > >
> > > [2] Parameter numbers extracted from the official publications.

---

> > > ### Author Response · Authors · 2025-12-03
> > > **Experimental Results - Added Paragraphs**
> > >
> > > We added the following paragraphs to the main text in Section 4 summarizing all these tables:
> > >
> > > > While the previous comparisons demonstrate that mGRADE attains comparable accuracy with substantially fewer parameters, an important complementary question is how the architectures behave under iso-parameter conditions.
> > > To assess this, we optimize an LRU model matched to mGRADE in both parameter count and activation footprint. Under this iso-capacity setting, mGRADE achieves an average 9% higher accuracy across all LRA tasks, indicating a clear architectural advantage beyond parameter efficiency alone. Note that unlike the results reported in Orvieto et al. (2023), our LRU and mGRADE implementations are fully causal for every task, thus suitable for edge deployment.
> > >
> > > > In Table 4, we present the raw-speech classification results on Google Speech Commands, comparing to current state-of-the-art recurrent architectures. mGRADE attains an accuracy within 2% of Liquid-S4 while requiring 10% fewer parameters.
> > > In addition, mGRADE relies solely on real-valued operations, avoiding the complex-valued arithmetic used in Liquid-S4.
> > > This combination of reduced parameter count and simpler computation makes mGRADE a suitable candidate for deployment under hardware or energy constraints.
> > >
> > > > These results confirm that our proposed architecture is capable of tackling large-scale and long-range tasks, particularly for time-series data, thus validating our theoretical predictions and demonstrating clear advantages in memory footprint and performance at comparable network sizes. We also assess whether these advantages carry over when comparing against architectures tailored for edge deployment, such as spiking neural networks. As shown in the appendix D, mGRADE remains competitive in this regime, achieving accuracy within 2.6% of the strongest spiking model while requiring 3× fewer parameters.

---

### Official Review · Reviewer_7yyH · 2025-10-31

**Soundness:** 2
**Presentation:** 3
**Contribution:** 2
**Rating:** 4
**Confidence:** 4

**Summary:**

The paper introduces mGRADE which is a minimal linear GRU with convolutions that support parallel training using scan algorithm and RNN-style inference of State Space Models and Linear Transformers. The main motivation behind design is to build a device for embedded systems and resource constraint environments. The experiments are done in LRA benchmark and Lorenz attractor.  moreover, the paper highlights that mGRADE enjoys selective gating which helps for long-context modeling (which is shown in LRA benchmark).

**Strengths:**

The major strengths of paper are:

- **Direction of the paper** Paper build upon a current important direction of efficient sequence modeling with sub-quadratic models such as linear transformers and SSMs.

- **Background analysis** Paper nicely present a very sufficient background material on SSMs and progress of literature specially in its related work section and introduction.

- **Loranz and LRA Experiments** The design of these experiments are indeed relevant and sound.

- **Presentation and writing** The paper is well-written and clearly presented its approach.

**Weaknesses:**

However, while the paper is generally well written and the overall design appears sound, there are several important gray areas regarding comparisons with SSMs—particularly Mamba, Mamba2, and GLA [2], as well as some outdated training strategies.

* **Model differences with Mamba2 and Gated RFA:**
  The main architectural components appear very similar to existing SSMs. In particular, the sequence mixing mechanism based on the minimal GRU closely resembles the recurrence structure of the Gated RFA. Moreover, the role and necessity of short-range causal Conv1D layers have already been explored in modern Linear Transformers such as Gated DeltaNet [1] and in prior analyses (see [this post](https://kexue.fm/archives/11320)). This raises the question: what is the key difference and contribution of mGRADE compared to existing models, given that both causal Conv1D and GRU-style recurrence have been well studied in SSMs?

* **Lack of time-series experiments:**
  Although the paper claims to target sensory data and signal processing, it does not include any experiments on time-series tasks. The only experimental setup is the LRA benchmark, which is both outdated and largely synthetic. For a model motivated by real-world signals, it would be important to demonstrate performance on datasets such as *Speech Commands* (as used in S4 models), or other representative time-series benchmarks.

* **Training efficiency and scan algorithm:**
  The paper relies on the parallel scan algorithm for training, which has been shown to be significantly slower and less efficient than chunkwise parallel training strategies such as those used in SSD and GLA (see Figure 6 of the paper). It would be useful to clarify how the training time of mGRADE compares to Mamba2 and GLA, and whether mGRADE also supports chunkwise parallel training.

* **Selectivity and language modeling:**
  If the model is designed to be selective, how does it perform on language modeling tasks? Selective SSMs like Mamba were specifically developed for language tasks and are known to perform poorly on LRA benchmarks while excelling on linguistic datasets (as highlighted by the authors [here](https://github.com/state-spaces/mamba/issues/282#issuecomment-2221135197)). It would be important to see how mGRADE performs in comparison, and whether its selectivity mechanism aligns with those used in language-oriented SSMs.




-------


### References

[1] Gated Delta Networks: Improving Mamba2 with Delta Rule: Songlin Yang, Jan Kautz, Ali Hatamizadeh


[2] Gated Linear Attention Transformers with Hardware-Efficient Training: Songlin Yang, Bailin Wang, Yikang Shen, Rameswar Panda, Yoon Kim

**Questions:**

1) What is the training time of mGRADE compared to Mamba2 specifically since it is using SSD?

2) How does the model perform in time-seires tasks such as **Table 13** of S4 paper [1]?

3) How does mGRADE works without causal convoulution on LRA to see how performant is the sequence mixing of the mGRADE?

4) What is the difference of mGRADE's minimal GRU compared to GRFA's or Mamba2's sequence mixing?

------

### References

[1] Efficiently Modeling Long Sequences with Structured State Spaces: Albert Gu, Karan Goel, and Christopher R´e

---

> ### Author Response · Authors · 2025-11-20
> **Reply - Part 1**
>
> We thank the reviewer for their careful assessment of our submission and for the constructive feedback. After reviewing the comments, we realized that we may not have sufficiently highlighted two central contributions of our work (on top of the memory efficiency gains):
>
> (i) the temporal convolution in mGRADE incorporates learnable spacings, which is fundamentally different from the standard short 1D-convolutions used in many sequence model backbones. This mechanism enables the model to implement a flexible, learnable delay embedding, connecting directly to classical dynamical systems theory (e.g., Takens’ theorem) and providing formal guarantees about its ability to capture short-range temporal structure. In contrast to the dense 1D-convolutions typically used in SSM backbones, our learnable-spacing convolution directly modulates the temporal positions of inputs. In particular, it transforms the classical convolution operation into a dynamic delay operator that directly learns the appropriate temporal geometry of the data by moving kernel elements in time rather than only modifying weights at the respective kernel positions. In addition to aiding learning by providing an inductive bias for temporal structure in the data, this training method for the temporal convolution is also more parameter efficient by employing the temporal positions as parameters rather than the full sets of weights in dense kernels.
>
> (ii) In combination with the minimal gated recurrent block, mGRADE explicitly separates short-term (delay embedding) and long-term (gated recurrence) dynamics, promoting functional specialization across modules and yielding a model whose internal computation is more interpretable than standard sequence models.
>
> These aspects, i.e., the novelty of the learnable-spacing temporal convolution and the resulting interpretability benefits, were not commented on by the reviewers, and we acknowledge that we did not emphasize them clearly enough in the original submission. We have strengthened this discussion in the revised version, and in this light we address your specific concerns in detail below.
>
> > Weakness 1: ... what is the key difference and contribution of mGRADE compared to existing models, given that both causal Conv1D and GRU-style recurrence have been well studied in SSMs?
>
> Thank you for giving us this opportunity to clarify what we consider the main contributions of mGRADE compared to existing models. As you note and as we highlight in Section 5, Conv1D and minimal GRU-style recurrence have already been applied in other models, including modern Linear Transformers like Gated RFA, Gated DeltaNet, as well as SSMs like the Mamba models. Importantly, this demonstrates the strength of Conv1D and minGRU-style recurrence as inductive biases. Besides the fact that mGRADE is the most memory-efficient model able to handle the long-range dependencies of LRA, the key difference between mGRADE and other sequence models is that we use learnable spacings between kernel elements. Additionally, we add to this prior work by investigating why (not just if) the two architectural patterns of gated recurrence and temporal convolution complement each other.
>
> The use of learnable spacings between kernel elements is key to the convolution component’s theoretical equivalence to a delay embedding in the sense of Takens’ theorem (see Section 3.1). This helps mGRADE predict evolving and partially observed dynamics over short time frames. The learnable spacings also enable us to use convolutional kernels that extend over several timesteps while using only very few parameters. Finally (and very interestingly), the pattern of spacings that emerge in the kernel during learning reflects back the structure of the task itself. For example, Figure A7 shows how the kernel elements space out in 1D space to be local in 2D space for the Image task (despite the image being presented as a 1D sequence) while for the ListOps task a temporal hierarchy across layers emerges matching the nested dependencies in ListOps. We have updated the main text to better highlight the centrality of the learnable spacings in mGRADE.
>
> Of course, the gated recurrent component is also important. Given that it is a common architectural pattern in recurrent models (like Gated RFA and many others), we focused on highlighting why it is a useful component for long-range dependencies (complementary to the blog post mentioned by the reviewer). Building on theoretical insights developed in [1] and [2], we show how the gated recurrent architecture complements the learnable spacings by enabling selective storage of long-term information (see Section 3.2). The convolution with learnable spacings on the other hand effectively counteracts the low-frequency bias of the gated recurrent component without impairing its selectivity (see Appendix B.3).
>
> \[continues in Part 2\]

---

> ### Author Response · Authors · 2025-11-20
> **Reply - Part 2**
>
> Thus, we believe this paper extends beyond Mamba-2, Gated RFA, or other comparable SSMs, by building a theoretical foundation for why combining 1D convolutions and gated recurrences has proven to be a useful inductive bias while also presenting a novel recurrent sequence model that incorporates learnable spacings between kernel elements. As far as we know, the use of DCLS for learnable spacings in a sequence model has so far only been demonstrated in the context of spiking neural networks [3], with this paper representing the first instance that these are integrated with a gated minGRU-style recurrence and successfully evaluated on long-range dependencies.
>
> > Weakness 2: Lack of time-series experiments...
>
> We thank the reviewer for this suggestion and agree that it would greatly improve the body of our work. Validating our claims by showing empirical results on time-series benchmarks is now our priority, and we are actively working on it. We will post a new table with the results in the coming days.
>
> > Weakness 3: ...clarify how the training time of mGRADE compares to Mamba2 and GLA, and whether mGRADE also supports chunkwise parallel training.
>
> Our primary design goal is a memory-efficient sequence model for edge and resource-constrained settings. The wall-clock advantages of more sophisticated scan algorithms such as SSD typically appear at much larger state sizes and sequence lengths \(T\) (e.g. large language models) than the regimes we target here.
>
> That said, we thank the reviewer for this suggestion and we fully agree that SSD-style training could be useful for future, larger mGRADE variants. Mamba-2 explicitly defines the scalar SSM scan (“cumprodsum'') as (see [4], Section B.1):
>
>
> *"The scalar SSM scan is defined as $h_t = a_t h_{t-1} + b_t $. Here $h_{-1}$ can be an arbitrary value representing the previous hidden state to the SSM Recurrence; unless otherwise specified, we assume $h_{-1} = 0$. We also call [this] equation the cumprodsum (cumulative product sum)."*
>
> To address the cost of evaluating this scan, they introduce a state-passing (chunkwise) mode (see [4], Section B.3.1):
>
> *"This mode can be viewed as a generalization of the standard recurrent mode where instead of passing forward the recurrent state $h$ one step at a time, we compute the answer on chunks of arbitrary length $Q$ and pass the state through the chunk."*
>
> Comparing this directly to mGRADE's minGRU update,
>
> $$
> h_t^{(i)} = \bigl(1 - z_t^{(i)}\bigr)\,h_{t-1}^{(i)} + z_t^{(i)}\,\tilde h_t^{(i)},
> $$
>
> we can identify $a_t^{(i)} = 1 - z_t^{(i)}$ and $b_t^{(i)} = z_t^{(i)} \tilde h_t^{(i)}$, so each dimension $i$ is exactly a scalar SSM scan. An SSD implementation would therefore reduce the effective scan length (and hence the span of a parallel prefix-scan) from $T$ to $T/Q$ by scanning only over $T / Q$ chunk-final states (for chunk size $Q$), changing the span from $O(\log T)$ to $O(\log(T / Q))$ while keeping the total work in $T$ essentially unchanged and modestly increasing activation memory due to chunk buffers. This chunkwise training scheme could be explored in future versions of mGRADE for large language applications.
>
> > Weakness 4: If the model is designed to be selective, how does it perform on language modeling tasks?
>
> While we agree that selectivity was incorporated into Mamba specifically to achieve higher performance on language modeling tasks (and to compete with alternative large language model architectures), our primary focus lies on tasks that would need to be solved in real time at the edge with limited compute and memory. This includes language classification tasks such as the Text and Retrieval tasks in LRA, visual tracking tasks such as the Image or Pathfinder tasks in LRA, and timeseries processing such as the Lorenz regression task. Even though none of these tasks involve autoregressive language modeling, they still require compressing a long history of inputs into a fixed size hidden state, which mGRADE’s selective gating achieves.
>
> Nevertheless, mGRADE’s strong performance on the Flip-Flop and Selective Copying tasks, which synthetically mimic many patterns found in natural language (for details and references see Section 3.2, particularly the fifth paragraph), indicates that its selectivity mechanism aligns with those used by language-oriented SSMs. This hints at potential language modeling capabilities, but we leave that exploration (possibly requiring an extension to matrix states, see HGRN2 [5]) to future work, choosing instead to concentrate on the underexplored problem of memory-efficient edge processing.
>
> > Question 1: What is the training time of mGRADE compared to Mamba2 specifically since it is using SSD?
>
> We refer the reviewer to our answer to weakness #3.
>
> > Question 2: How does the model perform in time-seires tasks such as Table 13 of S4 paper [1]?
>
> We refer the reviewer to our answer to weakness #2.

---

> ### Author Response · Authors · 2025-11-20
> **Reply - Part 3 (+ references)**
>
> > Question 3: How does mGRADE works without causal convolution on LRA to see how performant is the sequence mixing of the mGRADE?
>
> We agree that examining the core sequence mixing capabilities of mGRADE is essential. The model without the causal convolution component reduces to the minGRU architecture. We have performed this important ablation study and refer the reviewer to Table A7, where the results are presented. The data shows that while minGRU achieves approximately on par performance on ListOps, its performance significantly lags behind mGRADE on the Image task and fails on Pathfinder, demonstrating the critical role of the causal convolution for robust sequence mixing across diverse modalities. We highlight this in the main text (last paragraph of Section 4).
>
> > Question 4: What is the difference of mGRADE's minimal GRU compared to GRFA's or Mamba2's sequence mixing?
>
> We refer the reviewer to our answer to weakness #1.
>
> ### References
>
> [1] A. Gu and T. Dao. Mamba: Linear-time sequence modeling with selective state spaces. In First Conference on Language Modeling, 2024
>
> [2] Y. Sarrof et al. The expressive capacity of state space models: A formal language perspective. Neurips, 2024
>
> [3] I. Hammouamri et al. Learning delays in spiking neural networks using dilated convolutions with learnable spacings. ICLR, 2024
>
> [4] T. Dao and A. Gu. Transformers are SSMs: Generalized models and efficient algorithms through structured state space duality. arXiv preprint arXiv:2405.21060, 2024
>
> [5] Z. Qin et al. Hgrn2: Gated linear rnns with state expansion. Conference on Language Modeling, 2024

---

> > ### Comment · Reviewer_7yyH · 2025-11-26
> >
> > Thanks for authors reply. However, some of my concerns are still remaining:
> >
> > As authors mentioned that "the temporal convolution in mGRADE incorporates learnable spacings, which is fundamentally different from the standard short 1D-convolutions used in many sequence model backbones." This is still not shown mathematically anywhere in the paper and in the figure still is shown as Conv1D with spacing. Moreover, as also highlighted in the appendix DCLS is introduced already at [1], which is not new for this paper an it is only used as design choice. Same also holds for the GRU as sequence mixer.
> >
> > Regarding Weakness 3, I thank the authors for the clarification. I am aware that SSD or any chunk-wise parallel form can be used for training mGRADE. However, without a chunk-wise formulation or a specialized kernel for fast training—especially for sequences up to 16k (e.g., PathX)—there is a lack of novelty in the recurrence, as it is essentially the same as GRU. The convolution component is also the same as in [1]. Moreover, there is no time-series analysis, which was one of the stated motivations of the paper. Therefore, I maintain my score.
> >
> >
> > Minor: H3 is not cited correctly the citation is for Mamba.
> >
> >
> >
> > ### References
> >
> > [1] Ramin Hasani, Mathias Lechner, Tsun-Hsuan Wang, Makram Chahine, Alexander Amini, and
> > Daniela Rus. Liquid structural state-space models. In The Eleventh International Conference on Learning Representations, 2023.

---

> > > ### Author Response · Authors · 2025-12-01
> > > **Additional clarification**
> > >
> > > > DCLS is already introduced in Hasani et al., 2023...
> > >
> > > DCLS was introduced in [1] (by Hassani et al.) for spatial CNNs (not [2] although we understand that the similarity between the first author's surnames may have caused the confusion). Thus, combining learnable spacings with the minimal GRU for sequence modeling is an entirely novel application of DCLS. As emphasized in our answer, the principal novelty of our work is a theoretical exploration and empirical evaluation of how mGRADE leverages the complementary strengths of these two components, similar to how HGRN combined input-dependent recurrent gating (already used in the GRU) and a decay rate that approaches 1 with depth (used in LRU and other SSMs).
> > >
> > > > This is still not shown mathematically anywhere in the paper...
> > >
> > > Given your and the other reviewer’s feedback, we moved the precise mathematical description of DCLS from the appendix to the main text (in Section 2). In addition, a thorough and novel mathematically founded explanation of why DCLS is so effective for sequence modeling can be found in Section 3.1, followed by a formally justified explanation for complementing it with the minGRU component. We have now emphasized the critical difference in Figure 1 between a standard Conv1D block and our 1D temporal convolution with learnable spacings.
> > >
> > > > Moreover, there is no time-series analysis...
> > >
> > > As mentioned in our answer, a time series analysis (of audio waveforms on the Google Speech Commands dataset) will be presented by the end of the week. In addition, Section 3.1 already contains an analysis of a time series regression task on the chaotic Lorenz attractor, showing that mGRADE outperforms sequence models without convolution components.
> > >
> > > > However, without a chunk-wise formulation or a specialized kernel...
> > >
> > > As far as we are aware, chunk-wise parallel training presents only few hardware-specific advantages when using transitions parameterized by element-wise vector products between vector-valued states (as mGRADE uses). Indeed, chunk-wise training was motivated primarily by concerns regarding the cubic time complexity of multiplying many dense transition matrices for parallel scan (which mGRADE avoids by using simple element-wise products between z and h), and the quadratic memory complexity of instantiating matrix-valued states for every step in the sequence (which we avoid by using vector-valued states, h). A nice explanation can be found here: [DeltaNet Explanation, Part 2](]https://sustcsonglin.github.io/blog/2024/deltanet-2/). Since we are not using matrix-valued transitions or states and are not targeting tasks with extremely long sequence lengths (as Mamba-2 does), we expect chunk-wise training to only marginally improve over parallel scan.
> > >
> > > > H3 is not cited correctly the citation is for Mamba
> > >
> > > Finally, we apologize for the confusion regarding the H3 results in Table 2. We in fact intentionally cite the Mamba paper in Table 2, because the H3 paper itself ([3] as stated in our main text) does not publish results on the selective copying task, but the authors of Mamba (which includes one of the authors of H3) provide results for the H3 model. We have updated the caption of Table 2 to further clarify this.
> > >
> > > ### References
> > >
> > > [1] Ismail Khalfaoui Hassani, Thomas Pellegrini, and Timothee Masquelier. Dilated convolution with learnable spacings. In The Eleventh International Conference on Learning Representations, 2023. URL https://openreview.net/forum?id=Q3-1vRh3HOA.
> > >
> > > [2] Ramin Hasani, Mathias Lechner, Tsun-Hsuan Wang, Makram Chahine, Alexander Amini, and Daniela Rus. Liquid structural state-space models. In The Eleventh International Conference on Learning Representations, 2023. URL https://openreview.net/forum?id=g4OTKRKfS7R.
> > >
> > > [3] Tri Dao, Daniel Y. Fu, Khaled Kamal Saab, Armin W. Thomas, Atri Rudra, and Christopher Re. Hungry hungry hippos: Towards language modeling with state space models. CoRR, abs/2212.14052, 2022. doi: 10.48550/arXiv.2212.14052. URL https://doi.org/10.48550/arXiv.2212.14052.

---

> > > ### Author Response · Authors · 2025-12-03
> > > **Experimental Results**
> > >
> > > To address the reviewer's concern about
> > >
> > > > Lack of time-series experiments [...] it would be important to demonstrate performance on datasets such as Speech Commands (as used in S4 models).
> > >
> > > We tackled the Speech Command dataset as suggested.
> > >
> > > | Model                                       | Parameters / Buff.      | Causal            | Bidirectional        |
> > > |---------------------------------------------|--------------------------|-------------------|-----------------------|
> > > | S4-LegS [1,2]                                | 307K / 49K               | 93.6              | 96.1                  |
> > > | S4-FouT [1,2]                                | 307K / 49K               | 91.8              | 95.3                  |
> > > | S4D-LegS [1,2]                               | 306K / 49K               | 93.6              | 95.8                  |
> > > | S4D-Inv [1,2]                                | 306K / 49K               | 93.4              | 96.2                  |
> > > | S4D-Lin [1,2]                                | 306K / 49K               | 93.4              | _96.3_                |
> > > | Liquid-S4 [1,2]                              | _224K_ / _5K_            | **96.8**      | -                     |
> > > | S5 [1,2]                                     | 280K / **1K**            | -                 | **96.5**              |
> > > | **mGRADE**                                   | **198K** / 20K           | _94.7_            | -                     |
> > >
> > > [1] Uses complex numbers in the recurrence.
> > > [2] Parameter numbers extracted from the official publications.
> > >
> > > We added the following to the main text:
> > > > we present the raw-speech classification results on Google Speech Commands, comparing to current state-of-the-art recurrent architectures. mGRADE attains an accuracy within 2% of Liquid-S4 while requiring 10% fewer parameters. In addition, mGRADE relies solely on real-valued operations, avoiding the complex-valued arithmetic used in Liquid-S4. This combination of reduced parameter count and simpler computation makes mGRADE a suitable candidate for deployment under hardware or energy constraints

---

### Official Review · Reviewer_UQK5 · 2025-10-31

**Soundness:** 3
**Presentation:** 3
**Contribution:** 2
**Rating:** 4
**Confidence:** 3

**Summary:**

The paper propose mGRADE, a new neural network architecture that combine a temporal convolution learned spacings with a gated recurrent component. The advantages of the method are demonstrated on two synthetic tasks, as well as on the Long Range Arena benchmark, where the number of parameters/parameter buffers is reduced by a factor of up to 8.

**Strengths:**

The idea of combining a recurrent gating structure with temporal convolutions is interesting to capture temporal patterns both over short and long time-scales.

The figures in the paper are overall good, for instance figure 1 gives great intuition about the method.

Several of the benchmarks are interesting (Lorentz, Flip-Flop), and are connected to the theory the authors develop.

**Weaknesses:**

The authors are focusing in the introduction about the need to create a memory efficient architecture. With this in mind, it would have been more interesting in the LRA experiments to compare the models with respect to runtime memory requirements of the task rather than the number of parameters as the difference between loading 1 million parameters and 1k parameters is tiny in terms of memory footprint. Hence saying that the memory footprint is reduced by a factor of 8 is misleading.

In your table 2, you report results for H3 on selective copying, but I am not seeing these results reported in the H3 paper. The H3 paper focus on induction head and associative recall which appears related but not the same. Can you please clarify where these numbers are found?

It would be more clear if the parameterization of the conv kernel is given in the main text rather than in the appendix
It would have been good if the authors clarified in the main paper how the model achieves a lower parameter count (i.e. is it by having fewer layers, fewer parameters per layer, fewer parameters in the convolutions etc.).
The flip-flop task is not described in sufficient detail

**Questions:**

In the appendix (line 778) you say that a temporal convolution needs a buffer to store past inputs with size proportional to the kernel length. Why this is the case when the convolution is computed with the FFT?

Why is Path-X (part of LRA) not included in the analysis? Did your model not work on this task?

---

> ### Author Response · Authors · 2025-11-20
> **Reply - Part 1**
>
> We thank the reviewer for their careful assessment of our submission and for the constructive feedback. After reviewing the comments, we realized that we may not have sufficiently highlighted two central contributions of our work (on top of the memory efficiency gains):
>
> (i) the temporal convolution in mGRADE incorporates learnable spacings, which is fundamentally different from the standard short 1D-convolutions used in many sequence model backbones. This mechanism enables the model to implement a flexible, learnable delay embedding, connecting directly to classical dynamical systems theory (e.g., Takens’ theorem) and providing formal guarantees about its ability to capture short-range temporal structure. In contrast to the dense 1D-convolutions typically used in SSM backbones, our learnable-spacing convolution directly modulates the temporal positions of inputs. In particular, it transforms the classical convolution operation into a dynamic delay operator that directly learns the appropriate temporal geometry of the data by moving kernel elements in time rather than only modifying weights at the respective kernel positions. In addition to aiding learning by providing an inductive bias for temporal structure in the data, this training method for the temporal convolution is also more parameter efficient by employing the temporal positions as parameters rather than the full sets of weights in dense kernels.
>
> (ii) In combination with the minimal gated recurrent block, mGRADE explicitly separates short-term (delay embedding) and long-term (gated recurrence) dynamics, promoting functional specialization across modules and yielding a model whose internal computation is more interpretable than standard sequence models.
>
> These aspects, i.e., the novelty of the learnable-spacing temporal convolution and the resulting interpretability benefits, were not commented on by the reviewers, and we acknowledge that we did not emphasize them clearly enough in the original submission. We have strengthened this discussion in the revised version, and in this light we address your specific concerns in detail below.
> >W1: ... the difference between loading 1 million parameters and 1k parameters is tiny in terms of memory footprint ...
>
> We agree that alongside the parameter count, the runtime memory requirements of a model are an important consideration, especially on edge devices. For this reason, we have calculated the runtime memory footprints of our model and the baseline models used in Table 3. This comparison can be found in Table A6.
>
> We also understand that, on general-purpose hardware such as GPUs or CPUs, the difference between storing 1M and 1k parameters may appear small relative to the total available memory, particularly when considering that an external storage medium is available. However, our target deployment setting is ASICs for embedded edge deployment, where the situation is fundamentally different: all model parameters must reside in on-chip memory (typically SRAM), and such devices often have only tens to hundreds of kilobytes of total memory available. We have included several examples of such devices in the table below, gathered from references [1]-[8]. In this context, both activation, routing, and parameter memory need to fit on this tiny system. Parameter memory can dominate the total footprint and can determine whether the model fits on the device at all. Thus, reducing parameters from ~1M to ~1k leads to a large practical gain in this setting.
>
> | | [1] | [2] | [3] | [4] | [5] | [6] | [7] | [8] |
> |:---:|:---:|:---:|:---:|:---:|:---:|:---:|:---:|:---:|
> | **Model Type** | LSTM+FFN | TCN | CNN | CNN | TCN+FFN | Spiking RNN | SNN | Delta-RNN |
> | **On-Chip Memory (for weights)** | 67kB | 132kB | 16kB | 349kB | 71kB | 138kB | 353kB | 48kB |
> | **\# Max. On-Chip Weights** | 32k | 400k | 32.8k | 2.1k | 133k | 132k | 194k | Depends on weight precision |
> | **Max. Weight Prec.** | 4bit | up to 8bit | 4bit | 16bit | 4bit | 8bit | 14bit | Configurable |
>
> *On-Chip Weight Memory refers to how many bytes are available for weights, routing, etc. (not including the activation buffer)*
>
> *# Max. On-Chip Weights refers to the actual maximum number of individual weight connections. This is separate from on-chip weight memory because many of these chips have hardwired network structures with fixed weight precision.*
>
> *Weight Precision refers to the highest possible weight precision.*
>
> > Weakness 2: In your table 2, you report results for H3
>
> The results of H3 we report are taken from the selective copying experiments in [9]. Based on your feedback, we annotated the corresponding row in table 2 to clarify this. Additionally, we expanded table 2 by adding the performance of TCNs on selective copy to enable a more comprehensive comparison.

---

> ### Author Response · Authors · 2025-11-20
> **Reply - Part 2**
>
> > Weakness 3: It would be more clear if the parameterization of the conv kernel ...
>
> Following your suggestion, we moved the parameterization of DCLS to the main text. We believe this also helps emphasize our novel use of learnable spacings in the convolution component of mGRADE. These learnable spacings enrich the model and serve as a useful inductive bias for sequence modeling tasks, which helps reduce the parameter count significantly.
>
> In fact, we believe these learnable spacings (equivalent to learnable delays) to be a key factor in reducing the number of parameters we use. Accordingly, we have modified the final two paragraphs in Section 3.1 to emphasize this point. In short, mGRADE can reconstruct partially observed dynamics thanks to its ability to express a delay embedding through its learnable spacings. This means that it can fully reconstruct the geometry of a partially observed dynamical system within its hidden state. Crucially, this reconstruction property only requires a number of parameters that scales with the dimensionality of the observed dynamical system and not the length of the observed sequence. Since the dynamical system’s dimensionality is fixed and usually smaller than the sequence length, this substantially decreases the number of parameters mGRADE requires relative to other recurrent models that need to store a complete history within their hidden state (e.g., Appendix A.2 “Why not parameterize the convolution as a SSM?” or the discussion about memory collisions in [10]). This aligns with previous work showing that learnable delays substantially increase the class of functions that can be approximated by a spiking neuron model [11] as well as the practical performance with less parameters [12].
>
> In terms of the actual architectural modifications made, we generally attempted to match the LRU and S4 in the number of layers while reducing the width of the network. Overall, the architecture for each task was chosen as described in Appendix B.1.2., B.3, B.4.2, B.5.1, and C.1. We added a sentence in Section 4 (“Experimental setup”) to highlight this in the main text as you recommended.
>
> Following your last suggestion, we have added a short paragraph describing the flip-flop task in more detail after the formal definitions in Section 3.2. In addition, Appendix B.4.2 includes significantly more details regarding the dataset size, sequence length, and probabilities of specific symbols occurring.
>
> > Question 1: ... Why this is the case when the convolution is computed with the FFT? ...
>
> The choice between using the FFT for convolution and the classical (direct) time-domain method is driven primarily by a trade-off between computational speed and memory overhead in different computing environments. While the FFT is theoretically much faster ($O(N \log N)$) than the direct method ($O(N \cdot \Gamma)$) for large inputs, making it ideal for GPU training where speed is paramount, the direct method is often favored for run-time inference on memory-constrained edge devices because of its dramatically lower memory footprint. Specifically, the direct method only needs to buffer the input for a duration equal to the small kernel length ($\Gamma$), using only real numbers. In contrast, even when using techniques like Overlap-Add to chunk long signals for the FFT method (avoiding the need for the entire signal), the memory requirements are still substantially higher. This is because the FFT output (and consequently, the processed kernel and input chunks in the frequency domain) must be stored as complex numbers (requiring both a real and an imaginary part), effectively doubling the storage needed and introducing more complicated processing logic compared to the simple multiply-accumulate operations of the direct method.

---

> ### Author Response · Authors · 2025-11-20
> **Reply - Part 3 (+ references)**
>
> > Question 2: Why is Path-X (part of LRA) not included in the analysis? Did your model not work on this task?
>
> We thank the reviewer (as well as Reviewer BrFx in MaW4) for pointing this out. We of course agree that having results on Path-X would strengthen our paper. However, we do not have access to a large amount of computational resources and therefore have to prioritize which tasks we will be tackling during this rebuttal process. We have assigned Path-X a lower priority compared to time series datasets for the following reasons:
>
> As far as we are aware, there exist very few, if any, prior results of causal, unidirectional, streamed architectures that publish results on this task. All architectures in Table 6 (SotA for LRA) employ bidirectional (i.e. acausal) inputs and therefore are not an appropriate comparison for our model (which is designed for causal and streamed input processing at the edge) on this task.
>
> The lack of prior results without bidirectionality hints to the fact that this task is extremely difficult for streaming and causal architectures.
>
> Runtimes of 36h for a single seed on our setups lead us to assign a low success chance for finding appropriate hyperparameters within a reasonable time for our model that would result in competitive performance on this task.
>
> ### References
>
> [1] J. S. P. Giraldo et al: A 65-nm speech-triggered wake-up soc for 10- µ w keyword spotting and speaker verification. In IEEE Journal of Solid-State Circuits 2020.
>
> [2] V. Jain et al: A tiny versatile system-on-chip with state-retentive mram for ml inference at the extreme edge. In IEEE Journal of Solid-State Circuits 2023.
>
> [3] F. Tan et al.  A 1.8% far, 2 ms decision latency, 1.73 nj/decision keywords-spotting (kws) chip incorporating transfer-computing speaker verification, hybrid-if-domain computing and scalable 5t-sram. In IEEE Journal of Solid-State Circuits 2025.
>
> [4] H. Yang et al. Fsl-hdnn: A 5.7 tops/w end-to-end few-shot learning classifier accelerator with feature extraction and hyperdimensional computing. ESSERC 2024
>
> [5] D. den Blanken and C. Frenkel. Chameleon: A MatMul-Free Temporal Convolutional Network Accelerator for End-to-End Few-Shot and Continual Learning from Sequential Data. In arXiv, 2025.
>
> [6] C. Frenkel and G. Indiveri. ReckOn: A 28nm Sub-mm2 Task-Agnostic Spiking Recurrent Neural Network Processor Enabling On-Chip Learning over Second-Long Timescales. ISSCC 2022
>
> [7] J. Park et al. A 65-nm 236.5 nJ/Classification Neuromorphic Processor with 7.5% Energy Overhead On-Chip Learning Using Direct Spike-Only Feedback. ISSCC 2019
>
> [8] S. Zhou et al. An 8.62-μW 75-dB DRSoC Fully Integrated SoC for Spoken Language Understanding. In IEEE Journal of Solid-State Circuits 2025
>
> [9] A. Gu and T. Dao. Mamba: Linear-time sequence modeling with selective state spaces. In First Conference on Language Modeling, 2024.
>
> [10] I. Schlag, et al.. Linear transformers are secretly fast weight programmers, ICML 2021.
>
> [11] W. Maass and M. Schmitt. On the Complexity of Learning for Spiking Neurons with Temporal Coding. Information and Computation, 1999
>
> [12] I. Hammouamri et al. Learning delays in spiking neural networks using dilated convolutions with learnable spacings. In The Twelfth International Conference on Learning Representations, 2024.

---

### Comment · Area_Chair_WhdG · 2025-11-24
**Discussion with Authors**

Dear Reviewers,

The authors have diligently provided responses to your questions and concerns. If you haven't already don so, I request you to please review the authors' responses, acknowledge that you have read them and actively engage with them in further discussion as needed.

This discussion period, with the authors, will end on December 2, 2025 (AoE). However, I request that you not wait until the last minute and actively engage with the authors early.

Best,
AC

---

### Author Response · Authors · 2025-12-03
**Last Words - Part 1**

Our submission makes three concrete, complete contributions, each supported by theory, ablations and experiment, (which we now have clarified explicitly in the introduction):

1. A learnable-spacing temporal convolution implementing a flexible delay embedding. We have shown that this mechanism has a formal grounding in classical dynamical systems theory, providing an explicit inductive bias for short-term structure, and we have validated this claim through multiple ablation studies.
2. A principled separation of short- and long-term dynamics via delay embedding and a minimal gated recurrent block. We have shown that this design leads to functional specialization and more interpretable internal computation than monolithic sequence models.
3. A thorough causal-streaming evaluation on LRA using minimal resources. Across tasks (and in the causal setting), we have shown that mGRADE achieves on-par or superior performance to the state-of-the-art baselines while using significantly fewer parameters. We have performed targeted ablations demonstrating the functional role of each component in our model architecture across time scales.

To strengthen our claims regarding these contributions, and to   address the reviewers’ concerns, we have added the following results:

1. Comparison to Parameter-matched LRU (on LRA task).  We trained a performance-optimized LRU constrained to the same parameter budget as mGRADE (see Table 3).   mGRADE significantly outperforms it, showing that our parameter efficiency cannot be reproduced by simply shrinking an existing SOTA model. This directly addresses the concern articulated by review BrFX: “how well, ,e.g, LRU would do with 40k parameters”.

2. Comparison  to other efficient architectures optimized for edge deployment  (on Spiking Heidelberg Digits (SHD) task). In response to reviewer BrFx’s suggestion to compare to models specifically designed for low-resource environments, such as spiking neural networks or neuromorphic architectures, we evaluated mGRADE’s performance on the spiking SHD benchmark, comparing to SotA spiking models (see Appendix D). Consistent with our claims, mGRADE achieves on-par or superior performance using a far smaller memory footprint. Thus, mGRADE compares very favourably to alternative edge processing models.

3. Evaluation on a time-series benchmark (Google Speech Commands (GSC) raw audio classification task). Following the recommendation by reviewer 7yyH and addressing the concern of reviewer BrFx regarding the lack of time-series experiments, we benchmarked mGRADE on raw-audio GSC (see Table 4). Here again, mGRADE achieves on-par or superior performance using a smaller memory footprint and simpler computations (without complex numbers in the transition).

Our original contributions along with these new additions make this paper a coherent and well-substantiated first milestone toward compact, interpretable sequence models with explicit multi-timescale structure, well within the scope of a ICLR submission.

We appreciate the reviewers’ perspective and thank them for recognizing the potential of this research direction.

---

> ### Author Response · Authors · 2025-12-03
> **Last Words - Part 2**
>
> To aid the AC in making their decision, we summarise the major weaknesses highlighted by the reviewers and our responses.
>
> Additional details can be found in the responses (as well as the updated paper).
>
> | Reviewer | Concern                                      | Response                                                                                                   |
> |----------|-----------------------------------------------|-------------------------------------------------------------------------------------------------------------|
> | UQK5     | Runtime Memory Requirements                   | Runtime requirements summarised in Table A7 (formerly A6)                                                   |
> | UQK5     | H3 results in Table 2                         | Clarified the origin of these results in main text (from Mamba paper)                                      |
> | UQK5     | Move parameterization of learnable spacings to main text | Done (see Section 2)                                                                              |
> | 7yyH     | Model Difference with Mamba2 and Gated RFA    | Neither uses learnable spacings and lack direct equivalence to delay embedding (clarified in Introduction) |
> | 7yyH     | Lack of time-series experiments               | Added evaluation on Google Speech Commands raw audio classification task (see Table 4)                     |
> | 7yyH     | Training efficiency and scan algorithm        | Chunk-wise training presents smaller advantages for small models with vector-valued states (see “Additional Clarification” response) |
> | 7yyH     | Selectivity and language modelling            | Smaller models are focused on edge processing tasks like audio processing (see GSC results in Table 4)     |
> | BrFx     | Comparing with performance-optimized SSMs     | Compared to performance-optimized SSM (LRU) that is parameter-matched to mGRADE (see updated Table 3)       |
> | BrFx     | Absence of efficient architectures            | Compared to models designed for edge processing (spiking neural networks) on SHD task (see Appendix D)                                   |
> | BrFx     | Absence of edge evaluation                    | mGRADE’s required memory footprint matches current ASIC sizes (see “Reply – Part 2” response to reviewer BrFx) |
> | BrFx     | Absence of PathX                               | Prioritised GSC raw audio classification task since it is more relevant to edge applications               |
> | BrFx     | Importance of long convolutions?              | Ablation study removing the convolutional component cannot match mGRADE’s final performance (see Appendix E.1) |

---

### Meta-Review · Area_Chair_CwRV · 2026-01-05

**Summary:**

The reviewers generally found the paper to be well written, clearly motivated, and exploring an important direction, namely lightweight sequence modeling for memory-constrained and edge settings. The idea of combining minimal recurrent gating with temporal convolutions, along with the use of learnable spacings, was viewed as interesting, and several reviewers appreciated the clarity of the architectural presentation, theoretical grounding on synthetic tasks, and competitive performance on selected Long Range Arena benchmarks.

However, the central claim of memory efficiency was not convincingly supported, which ultimately informed the negative recommendation. Multiple reviewers expressed concern that the paper equates parameter count reduction with practical memory efficiency, while runtime memory, buffering, activations, and deployment-level constraints are either insufficiently analyzed or deferred. One reviewer captured this concern particularly well using a Venn-diagram analogy, noting that the work sits in an uncomfortable middle ground: it compares against SSMs not optimized for memory, does not benchmark against architectures explicitly designed for low-resource or edge deployment, and does not include real edge or hardware-level evaluations. As a result, the paper does not clearly occupy the intersection of efficient architectures, fair baselines, and practical edge validation.

Additional concerns raised by the reviewers include:

- Incomplete and potentially unfair comparisons, especially the lack of parameter-matched or memory-optimized baselines.

- Limited evidence that the proposed components are fundamentally novel, with reviewers questioning how much the gains stem from known ingredients (minGRU-style recurrence and Conv1D variants) rather than a clearly differentiated architectural advance.

- Missing or deferred evaluations on challenging benchmarks (e.g., Path-X) and real time-series or deployment-relevant tasks, which weakens the generality of the claims.

- A gap between the stated motivation (edge and embedded systems) and the experimental evidence, which is largely GPU-based and does not demonstrate clear edge deployment feasibility.

While the reviewers acknowledged that the paper contains the core of a potentially strong contribution, they agreed that, in its current form, the evidence does not substantiate the headline claims around memory efficiency and edge suitability. Consequently, the consensus was that the work is not yet ready for acceptance and would benefit from a more tightly aligned evaluation, clearer positioning within the existing literature, and stronger empirical validation of its primary claims.

**Reviewer Concerns:**

**Concerns partially addressed by the rebuttal**: The rebuttal helped clarify the design motivation and architectural choices, particularly the rationale for combining minimal recurrent gating with delay convolutions and the role of learnable spacings. Reviewers’ questions about synthetic task behavior and long-range dependency handling were partially addressed through additional explanations and references to existing theoretical results. The authors also clarified some experimental setup details, improving readability and intent.

**Concerns that remain outstanding**: However, the reviewers’ core concern regarding memory efficiency remains unresolved. Despite the rebuttal’s emphasis on reduced parameter count, reviewers remain unconvinced that the proposed model delivers meaningful practical memory savings, as runtime memory usage, activation storage, buffering, and deployment-level constraints were not directly evaluated. The Venn-diagram critique raised by one reviewer, highlighting the lack of overlap between efficient architectures, fair memory-optimized baselines, and real-world edge validation, was not substantively addressed. In addition, concerns about baseline fairness and positioning remain. Reviewers noted the absence of parameter- or memory-matched baselines and limited comparison against architectures explicitly designed for low-resource or edge settings. Questions regarding novelty, particularly the extent to which the method differs from existing minimal-gating and convolutional sequence models, also persist. Finally, the gap between the edge-centric motivation and the GPU-based experimental evidence remains unclosed.

Overall, while the rebuttal improved clarity, it did not sufficiently address the central concerns about memory efficiency, evaluation rigor, and alignment between claims and evidence.

**Reviewer Scores:**

- Reviewer UQK5 (Score: 2 – Reject): This reviewer’s primary concern was that the paper’s memory-efficiency claims are misleading, emphasizing that reductions in parameter count do not equate to meaningful runtime memory savings. They explicitly argued that practical memory usage, buffering, and deployment-level constraints were not evaluated. As these concerns were not resolved by the rebuttal, this reviewer would be very unlikely to increase their score.

- Reviewer BrFx (Score: 2 – Reject): This reviewer articulated a central critique using the Venn-diagram analogy, arguing that the paper does not convincingly lie at the intersection of memory-efficient architectures, fair and appropriate baselines, and realistic deployment or edge evaluation. In a follow-up comment, they explicitly stated that they were not swayed by the authors’ responses. Their score would certainly remain unchanged.

- Reviewer 7yyH (Score: 4 – marginally below the acceptance threshold): This reviewer acknowledged the clarity of the architecture and found aspects of the approach interesting, but still raised concerns about overstated memory-efficiency claims, missing or unfair baselines, and the lack of concrete runtime or edge-level memory evaluation. While this reviewer may have engaged constructively in further discussion, without new experimental evidence addressing practical memory usage, they would likely retain their score rather than increase it to acceptance.

Overall, given that two reviewers expressed firm reject positions (score 2) and one reviewer remained marginal (score 4), and that the rebuttal did not resolve the central concern regarding practical memory efficiency, there is little indication that fuller discussion alone, without new results on memory efficiency, would have led to upward score changes or a different final decision.

---

### Decision · Program_Chairs · 2026-01-26

Reject